# Functional imaging of nine distinct neuronal populations under a miniscope in freely behaving animals

**Mary L Phillips[1,2]\*, Nicolai T Urban[1], Taddeo Salemi[1], Zhe Dong[3], Ryohei Yasuda[1]\***

[1]Max Planck Florida Institute for Neuroscience, Jupiter, United States; [2]ZEISS Research Microscopy Solutions, White Plains, United States; [3]MetaCell, Boston, United States

## eLife Assessment

The new development of Neuroplex, a pipeline that links projection-defined neuronal identity to in vivo calcium activity within the same animal, is an **important** contribution to the field of neuroscience and beyond. The strength of evidence is **convincing**.

**Abstract** Head-mounted miniscopes have enabled functional fluorescence imaging in freely moving animals. However, current technology is limited to recording at most two spectrally distinct fluorophores, severely restricting the number of identifiable cell types. Here, we introduce multiplexed neuronal imaging (Neuroplex), a pipeline combining miniscope $Ca^{2+}$ recordings with in vivo multiplexed confocal spectral imaging to distinguish nine projection-defined neuronal subtypes through the same GRIN lens. By co-registering defined neurons with fluorophore-specific spectral fingerprints via linear unmixing, we link projection-defined identities to behaviorally relevant neuronal activity. This approach overcomes spectral constraints of miniscopes, enabling circuit-level dissection of behavior in single animals.

**\*For correspondence:**
mary.phillips@zeiss.com (MLP);
ryohei.yasuda@mpfi.org (RY)

## Introduction

A central goal of systems neuroscience is to understand how cell-type-specific activity gives rise to behavior. Miniaturized head-mounted microscopes have enabled in vivo calcium imaging in freely moving animals, greatly enhancing our understanding of neural encoding of behavior and experience (*Resendez and Stuber, 2015*). In the past decade, these 'miniscopes' *Ghosh et al., 2011* have evolved rapidly, gaining features, such as wireless capability (*Barbera et al., 2019*) increased field of view (*Guo et al., 2021*; *Scott et al., 2018*), two-photon capability (*Helmchen et al., 2001*; *Zong et al., 2017*; *Zong et al., 2022*), multiple focal planes (*Skocek et al., 2018*), and the ability to optogenetically stimulate as well as record (*Stamatakis et al., 2021*). Despite these advancements, the size and weight constraints of head-mountable microscopes continue to limit the amount of information obtainable from any single experiment.

One major constraint is spectral capacity: most miniscopes can distinguish only one or two fluorophores. Physical limitations on internal optics and filter sets restrict excitation and emission capabilities. In addition, the gradient-index (GRIN) lens induces severe chromatic aberrations along the optical z-axis, shifting the focal plane based on wavelength, often beyond the miniscope's focusing range (*Aharoni and Hoogland, 2019*). As a result, the current state-of-the-art typically allows the use of GCaMP in the green channel and a second marker in the red wavelength domain (*Aharoni and Hoogland, 2019*).

**Video 1.** Simultaneous behavior and projection-identified calcium activity during social interaction. Video showing animal behavior during the social memory task alongside the corresponding miniscope calcium recording from the same session. Calcium-active ROIs are pseudocolored according to their Neuroplex-assigned fluorophore identity, indicating the corresponding projection-defined neuronal population. https://elifesciences.org/articles/110277/figures#video1

To analyze more than two neuronal subtypes, researchers have resorted to using different animals with replicable behavioral tasks across cohorts (*Kohl et al., 2018*; *Kingsbury et al., 2020*). While this approach can yield meaningful comparisons, it prevents simultaneous analysis of distinct cell types within the same circuit. Several groups have attempted to overcome this limitation using post-hoc immunohistochemistry or in situ profiling, but these approaches are labor-intensive and incompatible with longitudinal studies (*Langer and Helmchen, 2012*; *Kahan et al., 2021*; *Xu et al., 2020*; *Khan et al., 2018*).

To address these challenges, we developed Neuroplex, a flexible imaging and analysis pipeline that enables the identification of multiple projection-defined neuronal subtypes observed during in vivo calcium imaging in freely behaving animals. Neuroplex combines functional recordings acquired through a head-mounted miniscope with multiplexed spectral confocal imaging performed through the same implanted GRIN lens, allowing fluorophore identities to be assigned to functionally defined neurons within the same live animal. This is achieved by first registering the location of active neurons measured with the miniscope to the confocal image stack using a Python-based image registration workflow and then extracting multiplexed spectral data from those registered neuronal locations. A linear unmixing algorithm based on experimentally derived spectral fingerprints is applied to assign one or more fluorophore identities to each registered neuronal location.

We demonstrate this approach using nine spectrally distinct fluorophores delivered via retrograde AAVs to downstream targets of the medial prefrontal cortex (mPFC), enabling the identification of projection-defined pyramidal neuron populations in vivo (*Video 1*). Across animals, approximately 70% of functionally defined neurons could be assigned to one of the injected fluorophores, with minimal false positives and high robustness supported by simulation and experimental modeling. Because all imaging is performed through the implanted GRIN lens in vivo, Neuroplex is fully compatible with longitudinal designs, allowing users to assess fluorophore identity prior to behavioral testing, or to re-image the same cell populations across multiple time points. This method avoids the challenges inherent to post-hoc co-registration between in vivo and post-fixation images, and expands the utility of miniscope-based experiments for circuit-level dissection.

## Results
### Correcting for GRIN-induced chromatic aberration

Neuroplex enables simultaneous tracking of multiple neuronal subtypes in behaving animals through a three-step pipeline: (1) Head-mounted GRIN-lens based miniscope imaging of GCaMP activity during behavior; (2) in vivo multiplexed confocal spectral imaging through the same GRIN lens to capture fluorophore fingerprints; and (3) linear unmixing to assign projection-specific identities to functionally defined neurons (*Figure 1*). Since the system is strongly influenced by the optical characteristics of the GRIN lens, we first characterized the chromatic and geometric aberrations of the most commonly used lens types in the field: 1×4 mm silver-doped lenses, typically used for mPFC imaging, and 0.6×7 mm lenses, available in both silver-doped and lithium-doped glass. The 1×4 mm format is currently only manufactured in silver-doped glass, which has a higher numerical aperture (NA). Lithium-doped lenses are more frequently used in dual-color experiments due to their reduced chromatic dispersion. Multiple lenses of each type were tested, and inter-lens variability was found to be negligible.

GRIN lenses introduce optical aberrations, including axial chromatic shifts (*Lee and Yun, 2011*) and lateral astigmatism. We measured the optical aberrations of each GRIN lens using a fluorescent calibration slide containing a pattern of equally spaced rings (each 1.8 µm in diameter, spaced

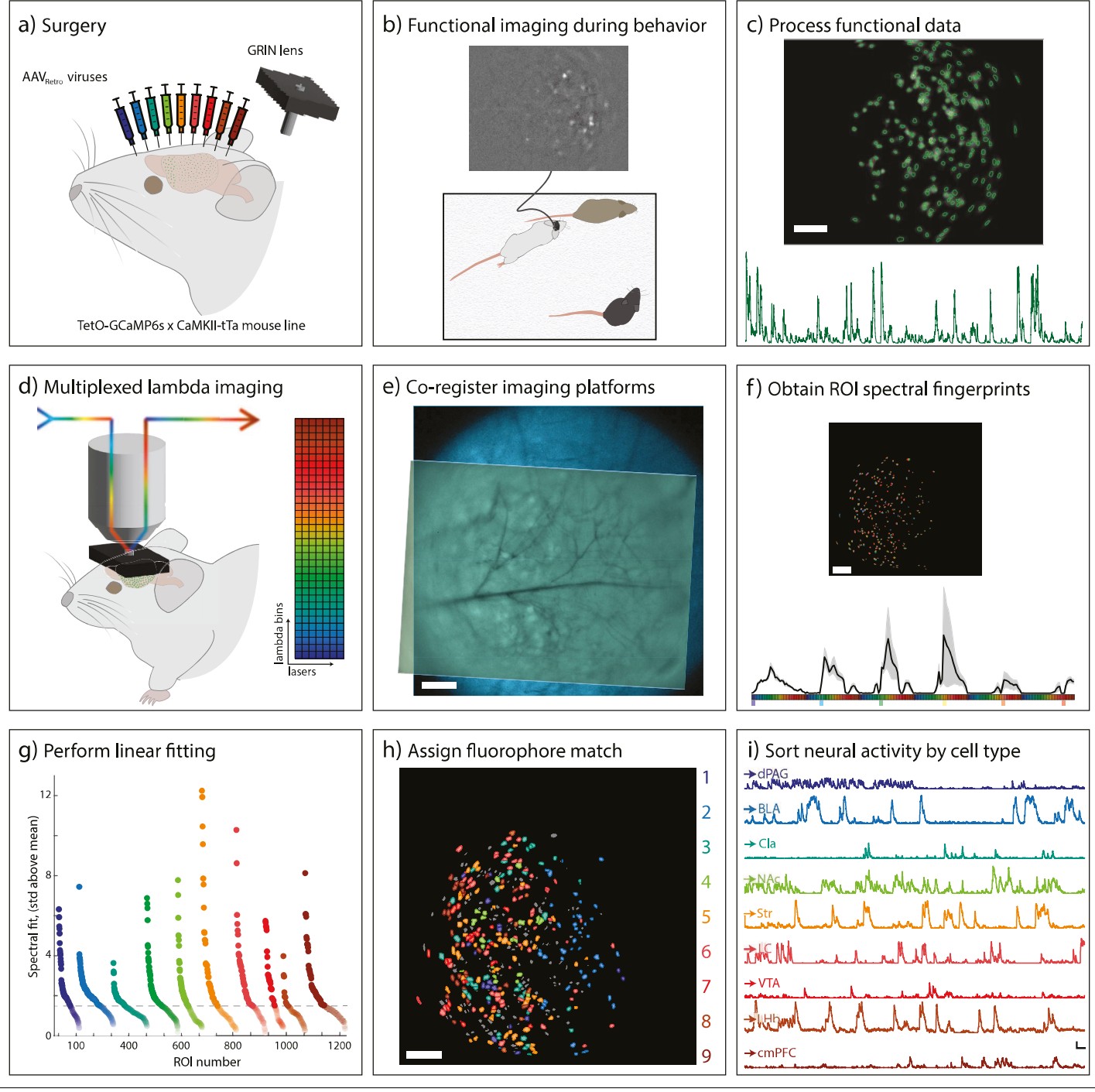

**Figure 1.** Experimental pipeline for identifying nine neuronal subtypes within behaviorally relevant regions of interest (ROIs) determined by GCaMP6s imaging. (**a**) Surgical paradigm. In a TetO-GCaMP6s × CaMKII-tTa mouse, 9 AAV_retro viruses are injected into downstream brain regions and gradient-index (GRIN) lens implanted into the target region. (**b**) Simultaneous recording of GCaMP6s (top) and behavior (bottom) during a social memory task. Scale bar = 100 μm (**c**) GCaMP6s recordings are processed. Constrained non-negative matrix factorization (CNMF)-defined ROIs (top) and ΔF/F traces (bottom) are exported. Scale bar = 100 μm. (**d**) Mice are head fixed and FOV under the GRIN lens imaged using the multiplexed lambda method. (**e**) Transformations are determined using anatomical background images to co-register the two imaging platforms. The transformations are applied to CNMF-defined ROIs. Scale bar = 100 μm. (**f**) Multispectral data are collected for each ROI (top) and an average spectral fingerprint for all ROIs is generated (bottom). Mean ±1.5 SD. Scale bar = 100 μm. (**g**) A linear unmixing model is applied to determine the fluorophore contribution for each ROI. Scale bar = 100 μm. (**i**) Neural activity is sorted by cell type. Scale bars = 20 ΔF/F (vertical), 20 s (horizontal).

50 μm apart) that were both excitable and emitted throughout the entire visible spectrum (*Royon and Converset, 2017*). A custom-made GRIN lens holder enabled precise and repeatable positioning above the calibration slide. We observed substantial chromatic aberrations along the optical z-axis, which increased in severity with the length of the GRIN lens (*Figure 2*). These aberrations caused a downward shift of the focal plane at longer wavelengths, which followed a second-order polynomial (*Figure 2a and b*, *Figure 2—figure supplement 1a*). The axial chromatic shift was significantly smaller for lithium-doped lenses as opposed to silver-doped lenses (*Figure 2—figure supplement 1b*). Chromatic aberrations in the lateral (x, y) plane were negligible across the field of view, including at the periphery (*Figure 2c*), indicating minimal off-axis distortion.

To overcome the axial chromatic aberrations during in vivo imaging, we adopted a two-step corrective strategy. First, we acquired spectral z-stacks spanning the full focal depth of the visible spectrum, using 405 nm excitation with 450 nm emission to define the upper bound and 639 nm excitation with 700 nm emission for the lower bound. These z-stacks were subsequently flattened to remove any wavelength-dependent focal shifts. Second, we widened the confocal pinhole to 350 μm to retain the longer-wavelength emission that would otherwise be excluded due to chromatic displacement from the nominal focal plane.

In addition to the axial chromatic aberrations, light transmission through GRIN lenses is also strongly wavelength dependent. Transmission peaked at 87% between 550–600 nm but declined sharply below 500 nm, falling to 58% at 405 nm (*Figure 2d*). This effect is most pronounced in silver-doped lenses and increased with lens length (*Figure 2—figure supplement 1c, d*). We fit a sixth-order polynomial to the transmission spectra to model wavelength-dependent scattering and used the resulting function to adjust excitation laser powers during confocal imaging: for in vivo experiments, we first identified which excitation laser induced the brightest emitted spectral bin. We then adjusted that laser's power and detector gain to achieve full dynamic range without saturation. The powers of all other lasers were subsequently scaled relative to this reference, using the polynomial-based transmission correction to ensure uniform illumination across wavelengths.

In addition to chromatic distortions, the GRIN lenses also induced lateral aberrations, particularly in the outer third of the FOV. We did not observe any directional field distortions (*Figure 2e*), but instead very prominent rotationally symmetric astigmatism as well as a strong curvature of the focal plane, both growing more severe with increasing distance from the center of the lens (*Figure 2f and g*). Despite their severity, these distortions were largely achromatic (*Figure 2c*), thus not requiring any chromatic corrections in the xy-plane.

## Fluorophore selection and optimization

Careful fluorophore selection is critical when spectrally distinguishing many different fluorophores. We analyzed published spectral profiles of available genetically encoded fluorophores and identified a combination of fluorescent proteins that could be uniquely separated by considering both their excitation and their emission spectra (*Lambert, 2019*). We determined that more fluorophores could be distinguished by employing a multiplexed-spectral imaging approach in which excitation lasers are sequentially activated and full emission spectra captured via spectral detectors. We then simulated the multiplexed spectral strategy in silico using various fluorophore combinations, before selecting ten fluorophores in addition to GCaMP6s for experimental validation. These included: mTagBFP2, mTurquoise2, T-Sapphire, mVenus, mPapaya, mOrange2, mScarlet, FusionRed, mCyRFP1, and mNeptune2.5. Each fluorophore was transfected individually into HEK293T cells, and its spectral fingerprint was recorded under multiplexed spectral imaging conditions for later use in linear unmixing (*Figure 2—figure supplement 2*). mPapaya was found to induce marked cell death in HEK293T cells and was, therefore, excluded from further study. The nine remaining fluorophores were used to make retrograde adeno-associated viruses (AAV$_{retro}$) for in vivo application.

## Identification of behaviorally relevant neurons

To test whether we could detect ten distinct fluorophores through GRIN lenses in behaviorally relevant neurons, we targeted mPFC projection neurons labeled via retrograde transport from up to nine downstream brain regions. Mice stably expressing GCaMP6s in pyramidal neurons (*Wekselblatt et al., 2016*) were injected with the nine different AAV$_{retro}$ viruses, each encoding a unique fluorophore, following two different fluorophore-region maps. Each map assigned a fluorophore to a brain region

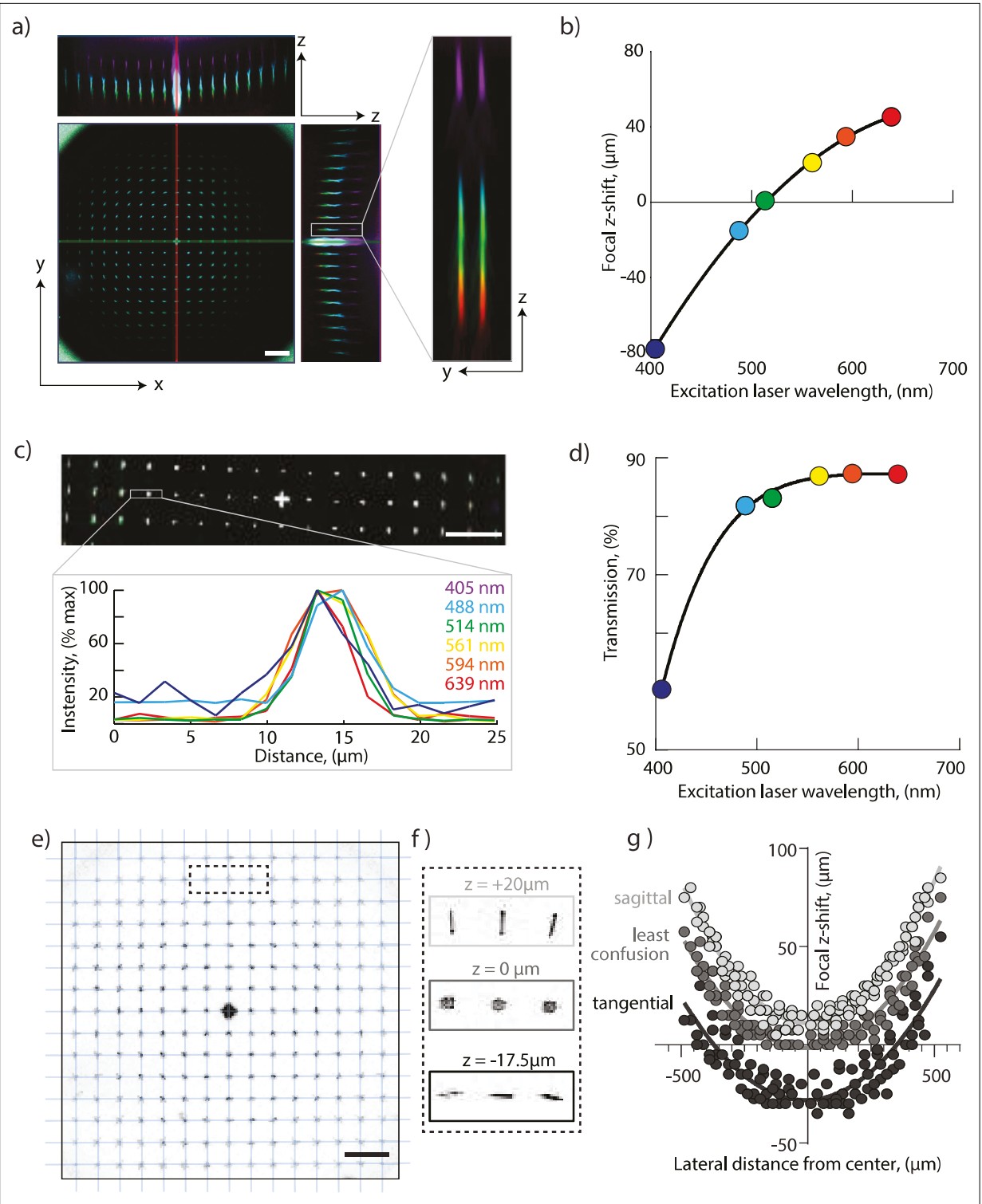

**Figure 2.** Gradient-index (GRIN) lens induced chromatic aberrations. (**a**) Multicolor image obtained through 1×4 mm silver-doped GRIN lens of the calibration slide highlighting the z-plane chromatic aberration. (**b**) Shift in z-focal plane as a function of excitation laser wavelength. Second-order polynomial $R^2$=0.9926, n=5. (**c**) Orthogonal projection of multicolor image obtained through 1×4 mm silver-doped GRIN lens of the calibration slide. Intensity profile (below) of single ring for each excitation channel shows negligible chromatic shift along lateral axes. (**d**) Percent transmission through the GRIN lens as a function of excitation laser wavelength. Sixth order polynomial $R^2$=0.9751, n=5. (**e**) Orthogonal projection of calibration slide imaged through 1×4 mm silver-doped GRIN lens overlaid (in cyan) with rectilinear grid lines. Substantial overlap of fluorescent rings from the grid indicates minimal field distortions. (**f**) Excerpt from (**e**) showing the rings focused in the sagittal plane (z=+20 μm), the circle of least confusion (z=0 μm), and the

*Figure 2 continued on next page*

*Figure 2 continued*

tangential focal plane (z=–17.5 μm). (**g**) Curvature of the Petzval field as a function of radial distance from center of the GRIN lens. Astigmatism results in three axially separated focal planes. Second order polynomial sagittal $R^2$=0.9845, least confusion $R^2$=0.8839, and tangential $R^2$=0.7519, n=3. Scale bars = 100 μm.

The online version of this article includes the following figure supplement(s) for figure 2:

**Figure supplement 1.** Comparison of aberrations across gradient-index (GRIN) lens types: Comparison of aberrations across GRIN lens types.

**Figure supplement 2.** Multiplexed spectral fingerprints for single-fluorophore samples.

known to receive projections from the mPFC. These regions included the dorsal periaqueductal gray (dPAG), basolateral amygdala (BLA), claustrum (Cla), nucleus accumbens (NAc), striatum (Str), locus coeruleus (LC), ventral tegmental area (VTA), lateral habenula (lHb), lateral hypothalamus (lHyp), and the contralateral prefrontal cortex (c-mPFC). AAV$_{retro}$ viruses are taken up by axon terminals and the encoded genetic sequences are transported back to the nucleus where the fluorophore is synthesized (*Tervo et al., 2016*).

During the same surgery, a 1×4 mm silver-doped GRIN lens with an integrated baseplate and head-bar was implanted directly above the prelimbic region of the mPFC (*Figure 1a*). After allowing five weeks for recovery and fluorophore expression, mice underwent a social memory task while GCaMP6s activity was recorded using a head-mounted miniscope (*Figure 1b*). The location of neurons active during this task was identified as regions of interest (ROIs) using a constrained non-negative matrix factorization algorithm (CNMF) (*Zhou et al., 2018*). The number of these behaviorally relevant ROIs ranged from 105 to 440 between subjects. These ROIs, along with a time-averaged fluorescence image of the field of view, were exported for co-registration and spectral analysis (*Figure 1c*).

Following functional imaging of GCaMP6, we performed spectral imaging using a confocal microscope to identify the fluorophore identity. To minimize the time-dependent changes of GCaMP emission during imaging, we silenced cortical activity by anesthetizing the animal with ketamine before head-fixing it under the confocal microscope to prevent motion. Using the parameters determined from GRIN lens characterization, multiplexed spectral z-stack images were acquired for the entire field of view under the GRIN lens (*Figures 1d, e and 3a*). A rolling ball background subtraction of 30 μm was applied to each z-plane to reduce neuropil interference, and the stack was then summed along the z-axis to counteract chromatic z-aberration. To co-register the two imaging modalities, we input the 512 nm emission channel from the 405 nm excitation laser, which provides strong vascular contrast and, therefore, serves as the confocal reference image, and the time-averaged GCaMP6s image from the behavioral session into a custom Python-based registration algorithm (*Figures 1f and 3b*). Using a two-step process, first a coarse and then a fine adjustment process, the code identified the optimal x-y shift and rotation by maximizing correlation coefficients. After applying these transformations to the GCaMP ROI masks (*Figure 3c*), we extracted multiplexed spectral data for each ROI by averaging the fluorescence from all pixels within the mask to generate a unique spectral profile (*Figure 3d*).

## Spectral unmixing of in vivo ROI fingerprints

Next, we unmixed the spectral data to identify which of the ten different fluorescent proteins were present in each ROI. To do so, we modeled the ROI spectral curve by using a linear unmixing algorithm, which determines the contribution of each fluorophore by best fitting a multiplier (beta) for each known spectral fingerprint (*Figure 1h*, *Figure 3e and f*) before summing the weighted spectral contributions. These reference spectra were derived empirically from HEK293T cells imaged using the same multiplexed spectral procedure.

After fitting beta multipliers for each ROI, the spectral baseline was calculated by averaging the beta multipliers for each fluorophore over all ROIs within the FOV. This per-subject normalization step accounts for animal-to-animal variability in background signal and expression levels, which can significantly alter baseline spectral profiles (*Figure 3—figure supplement 1*). A fluorophore hit was assigned if the beta value for a given fluorophore exceeded the mean baseline beta by more than 1.5 standard deviations. Representative examples of fluorophore-classified ROIs from in vivo experiments are shown (*Figure 3—figure supplement 2*), illustrating the confocal image, co-registered GCaMP ROI, extracted spectral fingerprint, and corresponding beta values for each of the nine fluorophores

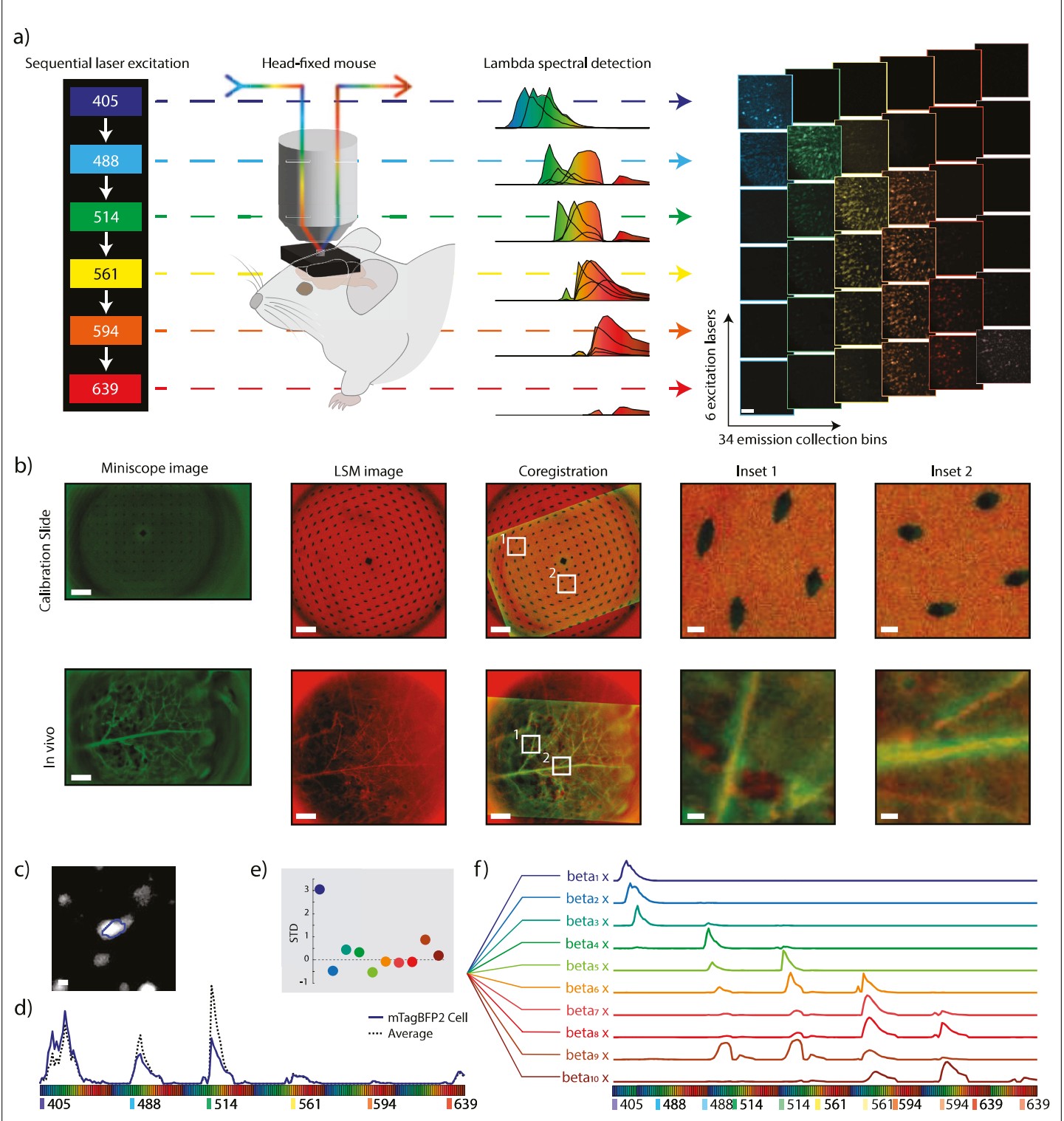

**Figure 3.** Identification of nine fluorophores through gradient-index (GRIN) lenses in vivo. (**a**) In vivo multiplexed spectral imaging paradigm. Schematic of multiplexed spectral imaging (left). Depiction of overlapping fluorophore spectral emissions for each excitation laser wavelength (middle). Depiction of multiplexed spectral images which create a 204-dimensional dataset (right). (**b**) Automated co-registration of miniscope and laser scanning confocal microscope (LSM) images. Top: A calibration slide used to measure scaling between modalities. Bottom: Experimental FOV showing brain vasculature. Miniscope and confocal images of the same FOV and automated co-registration overlay with zoomed-in regions of interest. (**c**) Example calcium-activity regions of interest (ROI) derived from miniscope data co-registered and overlaid on confocal LSM image. (**d**) Spectral fingerprint of the example ROI, with the solid blue line showing the example ROI and the dashed line depicting the average spectral profile of the animal. (**e**) Beta multiplier from the

*Figure 3 continued on next page*

*Figure 3 continued*

example ROI, depicting the deviation from the mean beta value for all ROIs from the same animal. (**f**) Empirically measured spectral profiles from pure fluorophore samples, shown as beta-weighted contributors to ROI fingerprints. Scale bar: 100 μm (**a**, **b**), 10 μm (**b** inset and **c**).

The online version of this article includes the following figure supplement(s) for figure 3:

**Figure supplement 1.** Subject-to-subject variation.

**Figure supplement 2.** Representative examples of fluorophore-identified neurons using Neuroplex.

**Figure supplement 3.** Modeling-based evaluation of fluorophore identification robustness under challenging conditions.

**Figure supplement 4.** Modeling dual fluorophore expressing regions of interests (ROIs).

used. This threshold was established through the simulation to balance sensitivity and specificity across subjects.

## Assessment of spectral unmixing approach

To determine the accuracy and robustness of our fluorophore identification algorithm component of Neuroplex, we validated the algorithm using a series of simulated datasets based on real spectral fingerprints. First, we expressed each fluorophore individually in HEK293T cells and collected spectral data from hundreds of cells per fluorophore. We then created test datasets by randomly combining spectral fingerprints in silico, allowing for any fluorophore composition. Additional perturbations were introduced, including increased GCaMP fluorescence (*Figure 3—figure supplement 3a, b*), simulated overlap of multiple fluorophores within the same ROI (*Figure 3—figure supplement 3c, d*), and white noise (*Figure 3—figure supplement 3e, f*).

In test datasets composed of equal numbers of single-fluorophore-expressing cells for all ten utilized fluorophores, the algorithm correctly estimated the beta contribution for each fluorophore to be approximately 10%, accurately reflecting the true distribution. Fluorophore identity was correctly assigned with nearly 100% accuracy across all fluorophores in the equal distribution dataset (*Figure 3—figure supplement 3g, h*). Under realistic experimental conditions, however, the distribution of fluorophores is unlikely to be equal and will vary between animals (*Figure 3—figure supplement 3i*). To model this, we tested datasets in which one fluorophore made up an increasing proportion of the total population. When a single-fluorophore exceeded 30% of all ROIs, the algorithm's ability to identify that fluorophore dropped sharply (*Figure 3—figure supplement 3g, h*), despite maintaining high accuracy for all other fluorophores. Upon closer inspection, weassigned most ROIs belonging to the over-represented fluorophores as false-negatives (i.e. no match; *Figure 3—figure supplement 3h*), as the fluorophore's spectral contribution deviated less from the spectral baseline.

To recover these false negatives without compromising the specificity of the cells expressing other fluorophores, we implemented a dual-pass identification strategy. ROIs that failed to reach the threshold for any fluorophore in the first pass were re-evaluated in a second pass, in which the identification threshold was dynamically adjusted based on the measured distribution of beta values for each fluorophore from the spectral baseline (*Figure 3—figure supplement 3j*). This dual-pass method successfully recovered over 90% of over-represented fluorophores, even when a single fluorophore accounted for 80% of the population, while retaining the unmodified high first-pass accuracy for the remaining fluorophores (*Figure 3—figure supplement 3g, h*).

We next tested the robustness of the unmixing algorithm under various imaging conditions. When increasing levels of GCaMP background were added to the spectra, identification accuracy declined gradually yet remained above 80% even when the GCaMP signal matched the fluorophore intensity (*Figure 3—figure supplement 3a, b*). Importantly, this perturbation resulted primarily in false negatives without increasing the rate of false positives, thereby maintaining specificity at the cost of efficiency. We then modeled the spatial overlap of cells expressing different fluorophores. As expected from our model, when the background contribution from a second fluorophore reached 50%, identification accuracy dropped to approximately 50%, again due mostly to false negatives (*Figure 3—figure supplement 3c, d*). Finally, we introduced increasing levels of Gaussian white noise. Although accuracy declined slowly under this condition, we observed a modest increase in false positives, reaching roughly 10% at a signal-to-noise (SNR) ratio of 4 (overall accuracy approximately 80%; *Figure 3—figure supplement 3f,*).

Finally, we estimated our fluorophore identification accuracy under realistic experimental conditions by simulating a dataset that mimicked key properties of an actual animal subject. For this dataset, we randomly sampled known fluorophore ROIs according to the measured fluorophore distribution across animals. We then added empirically measured background fluorescence and modeled GCaMP signal intensity at an average of 30% of the fluorophore brightness to reflect in vivo co-expression levels. To further challenge the algorithm, we introduced Gaussian white noise corresponding to a signal-to-noise ratio of 6 (*Figure 3—figure supplement 3k*). Under these conditions, the dual-pass method correctly identified 87% of all ROIs, with a false positive rate below 5%. Identification accuracy varied modestly across fluorophores, reflecting differences in spectral proximity and intrinsic brightness (*Figure 3—figure supplement 3l*).

These simulation-derived accuracy estimates characterize expected performance under defined noise and background conditions but were not formally propagated into confidence bounds on subtype proportions or behavioral comparisons. In this proof-of-principle study, subtype fractions are presented as assignment-dependent estimates rather than definitive anatomical measurements.

## Detection of two fluorophores within the same ROI

Until now, we had only attempted to identify a single fluorophore alongside the GCaMP signal. Excitatory neurons, however, often project to multiple downstream targets via *en passant* boutons or bifurcating axons, opening the possibility that some neurons may express more than one retrograde label. To assess whether Neuroplex could resolve such dual-expressing neurons, we simulated a dataset in which ROIs expressed every possible pairwise combination of two different fluorophores, alongside single-labeled controls (*Figure 3—figure supplement 4a*).

Overall, the algorithm was able to correctly identify at least one fluorophore in 98% of dual-expressing ROIs, and both fluorophores in 44% of the cases. As expected, performance varied by spectral separation: fluorophore pairs with high spectral similarity (e.g. mScarlet +FusionRed) were harder to resolve, while spectrally distinct pairs (e.g. mScarlet +mVenus) were classified more reliably. When identification errors occurred, they were primarily false negatives, with the algorithm failing to exceed the threshold for one of the fluorophores; false positives made up less than 1% of all errors.

Finally, we modeled the dual-expressing ROIs under realistic experimental conditions by adding empirically measured spectral background, GCaMP co-expression, and Gaussian white noise (*Figure 3—figure supplement 4b*). Under these conditions, the algorithm correctly identified at least one fluorophore in 91% of ROIs, and both fluorophores in 25% of dual-labeled ROIs. Among ROIs containing only one fluorophore, the algorithm falsely assigned a second fluorophore in 17% of cases. Across all test ROIs, the overall false positive rate was 14%, with 5% occurring in the primary fluorophore assignment and 9% in the secondary fluorophore assignment.

## Fluorophore distribution in behaviorally relevant ROIs

To independently assess whether the combination of projection target and fluorophore identity influenced the classification outcomes, we designed two complementary injection paradigms across five mice (*Figure 4a and b*). Each animal received the same set of nine AAVretro fluorophores, but in Group 2 the fluorophore-to-region pairings were rearranged to test detection robustness. Fluorophores with lower detection rates in Group 1 (*Figure 4a*) were reassigned to brain regions with higher labeling efficiency in Group 2 (*Figure 4b*).

For quantitative behavioral comparisons, each ROI was assigned a single primary fluorophore identity using a conservative winner-take-all rule. This assignment reflects the strongest spectral contribution and does not imply projection exclusivity. Rather, it provides a conservative lower-bound estimate of subtype proportions, as ROIs exceeding threshold for multiple fluorophores were classified according to their strongest spectral contribution.

Across animal subjects, Neuroplex assigned a fluorophore identity to 58–100% of behaviorally defined ROIs across five animals (*Figure 4c*). ROIs were extracted from calcium imaging using CNMF, such that only neurons with activity during the behavioral session were included in the dataset. This approach inherently excludes silent but anatomically labeled cells from analysis, and thus the classification rate reflects the proportion of functionally active neurons with identifiable fluorophores. Each animal had between 7 and 9 of the injected fluorophores successfully detected, with 1327 neurons

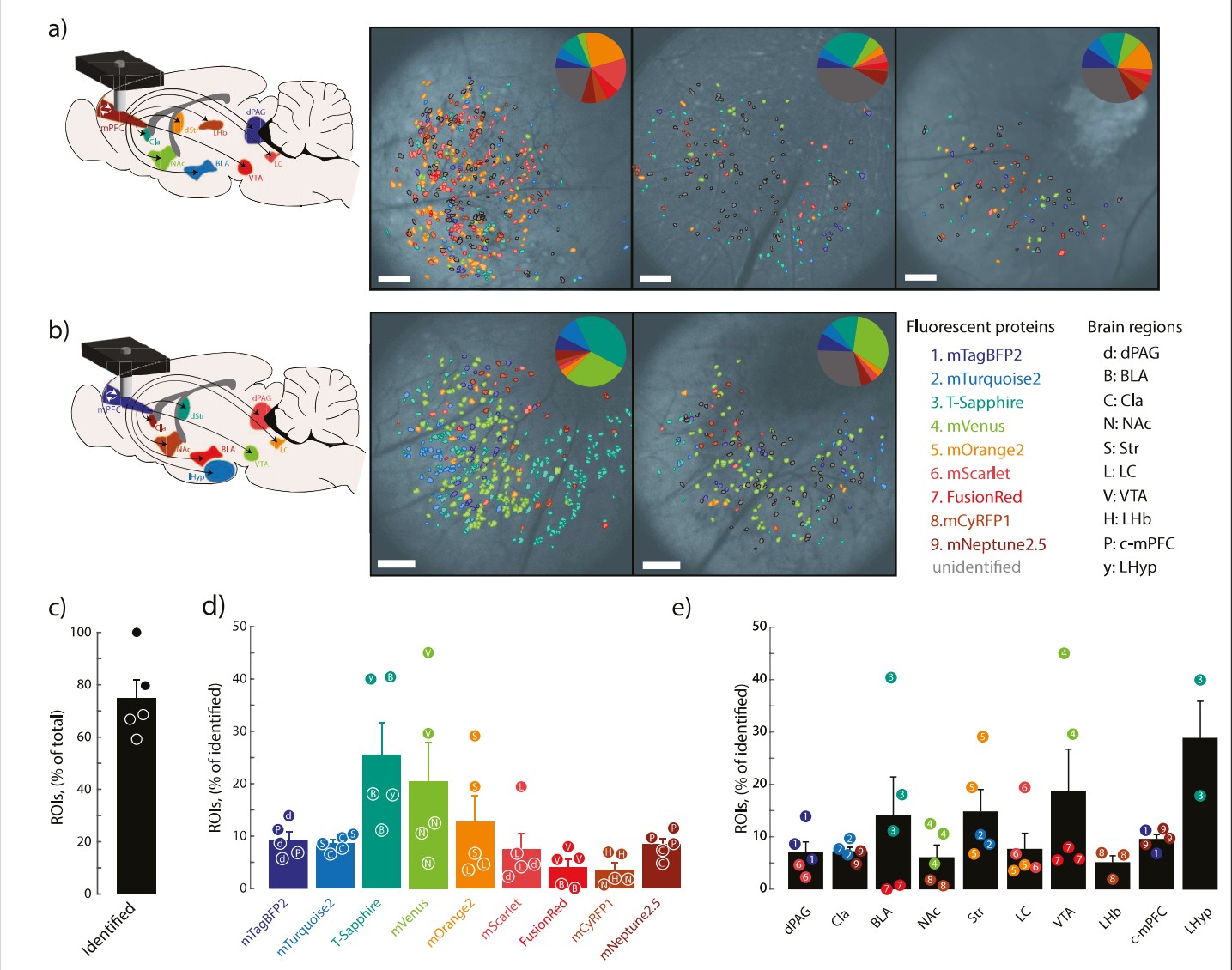

**Figure 4.** Distribution of fluorophore-positive functionally defined regions of interests (ROIs). (**a**) Identified fluorophores for injection paradigm A. Viral injection paradigm A consisted of mTagBFP2 into the dorsal periaqueductal gray (dPAG), mTurquoise2 into the basolateral amygdala (BLA), T-Sapphire into the claustrum (Cla), mVenus into the nucleus accumbens (NAc), mOrange2 into the striatum (Str), mScarlet into the locus coeruleus (LC), FusionRed into the ventral tegmental area (VTA), mCyRFP1 into the lateral habenula (LHb), and mNeptune2.5 into the contralateral prefrontal cortex (c-mPFC) (left). Spatial distribution of ROIs and respective fluorophore matches overlaid on anatomical images from the same mouse (right). Distribution of identified fluorophores per mouse (inset). (**b**) Identified fluorophores for injection paradigm B. Viral injection paradigm B consisted of mTagBFP2 into the c-mPFC, mTurquoise2 into the LHyp, T-Sapphire into the Str, mVenus into the VTA, mOrange2 into the LC, mScarlet into the dPAG, FusionRed into the BLA, mCyRFP1 into the NAc, and mNeptune2.5 into the Cla (left). Spatial distribution of ROIs and respective fluorophore assignments overlaid on the anatomical image from the same mouse (right). Distribution of identified fluorophores per mouse (inset). Color and letter codes for fluorophores and injection regions, respectively (far right). (**c**) Percentage of identified ROIs with a fluorophore match. Animal n=5; ROI n=1,327. (**d**) Percent of cells identified for each fluorophore. Letter insets on individual data points correspond to injected regions. N=1072. One-way ANOVA $p$=0.0071. (**e**) Percent of cells identified for each injected region. Number insets on individual data points correspond to injected fluorophore. N=1072. Mean ± SEM. One-way ANOVA, $p$=0.2599. Scale bars: 100 µm.

The online version of this article includes the following figure supplement(s) for figure 4:

**Figure supplement 1.** Experimental regions of interests (ROIs) expressing dual-fluorophores.

identified as active during behavior and 1156 of those being annotated with fluorophore identity (*Figure 4*, *Figure 3—figure supplement 1*).

Detection rates varied between fluorophores, and these differences were not solely attributable to projection target. For example, mVenus and T-Sapphire had the highest detection frequencies. While T-Sapphire was consistently associated with regions exhibiting high expression across both groups, mVenus was preferentially detected compared to the other fluorophore co-injected into the same region. In contrast, FusionRed and mCyRFP1 were identified least frequently, despite being assigned to brain regions where their co-injected fluorophore was readily detected (*Figure 4d*). This likely reflects lower brightness or expression of the fluorophores themselves and reflects predictions from in silico modeling (*Figure 3—figure supplement 3*).

Projection-defined detection patterns also emerged. Neurons projecting to the claustrum (Cla), striatum (Str), and ventral tegmental area (VTA) were detected most frequently, consistent with known dense innervation from the prelimbic cortex (PL). Conversely, neurons projecting to the contralateral mPFC (c-mPFC), lateral habenula (lHb), and dorsal periaqueductal gray (dPAG) were detected less often. (*Figure 4e*, *Figure 3—figure supplement 1*).

We next assessed how often functionally defined ROIs exceeded threshold for more than one fluorophore. Forty percent of ROIs had beta multipliers above threshold for a second fluorophore (*Figure 4—figure supplement 1a*). These secondary hits were most common among fluorophores at the edges of the spectrum, including mTagBFP2, mTurquoise2, mCyRFP1, and mNeptune2.5 (*Figure 4—figure supplement 1b*). Under high-background conditions, these fluorophores may be more permissive to co-occurring signals due to reduced spectral interference with other fluorophores. When grouped by brain region, dPAG, c-mPFC, and BLA-projecting neurons showed the highest rates of secondary fluorophore assignment (*Figure 4—figure supplement 1c*). The distribution of double-labeled ROIs, however, did not align with known patterns of projection convergence from prior anatomical studies (*Gongwer et al., 2022*). This discrepancy likely reflects differential fluorophore expression and variable discriminability of fluorophore combinations. For studies where precise identification of multi-labeled neurons is critical, we recommend pairing spectrally distinct fluorophores for regions with known or expected projection overlap during experimental design. Representative in vivo examples of neurons with two spectrally distant fluorophores above threshold (ROI 28), two more similar fluorophores above threshold (ROI 98), and a neuron with one fluorophore above and a second just under the detection threshold (ROI 142) are shown (*Figure 4—figure supplement 1d, e, f*).

Although dual-threshold ROIs were detected in vivo, these secondary assignments were not incorporated as co-identities in the primary behavioral analyses. This decision reflects a conservative specificity-first framework designed to minimize false-positive multi-label calls under realistic noise conditions. Accordingly, dual-label rates reported here should be interpreted descriptively. The present study focuses on demonstrating the feasibility of projection-resolved stratification, rather than providing definitive quantification of projection convergence.

## Neuronal cell types and behavior

We applied our Neuroplex pipeline for a social memory assay. In this assay, animals were trained to recognize a familiar conspecific, followed by a test session where both the familiar and a novel mouse were introduced into the arena (*Figure 5*, *Figure 5—figure supplement 1a*). We extracted spatial and temporal components of neuronal activity from miniscope videos using constrained non-negative matrix factorization for endoscopic data (CNMF-E) implemented through the Inscopix Data Processing Software. This approach identified spatial footprints and corresponding calcium traces for functionally active neurons during behavior. We then stratified these neurons by projection target and examined behaviorally selective activity across cell types. These analyses were performed using conservative single-label assignments; dual-threshold ROIs were not treated as co-identities in order to avoid overinterpretation of potentially ambiguous multi-label cells. Because identity assignment prioritizes specificity and classification uncertainty was not formally propagated into downstream comparisons, subtype fractions and behavior-by-subtype differences should be interpreted as qualitative demonstrations of projection-resolved functional stratification rather than precise anatomical quantifications.

For example, NAc-projecting neurons were selective to social interactions with either familiar or novel conspecifics (*Figure 5a and c*). In contrast, LC-projecting neurons were selective for aggressive

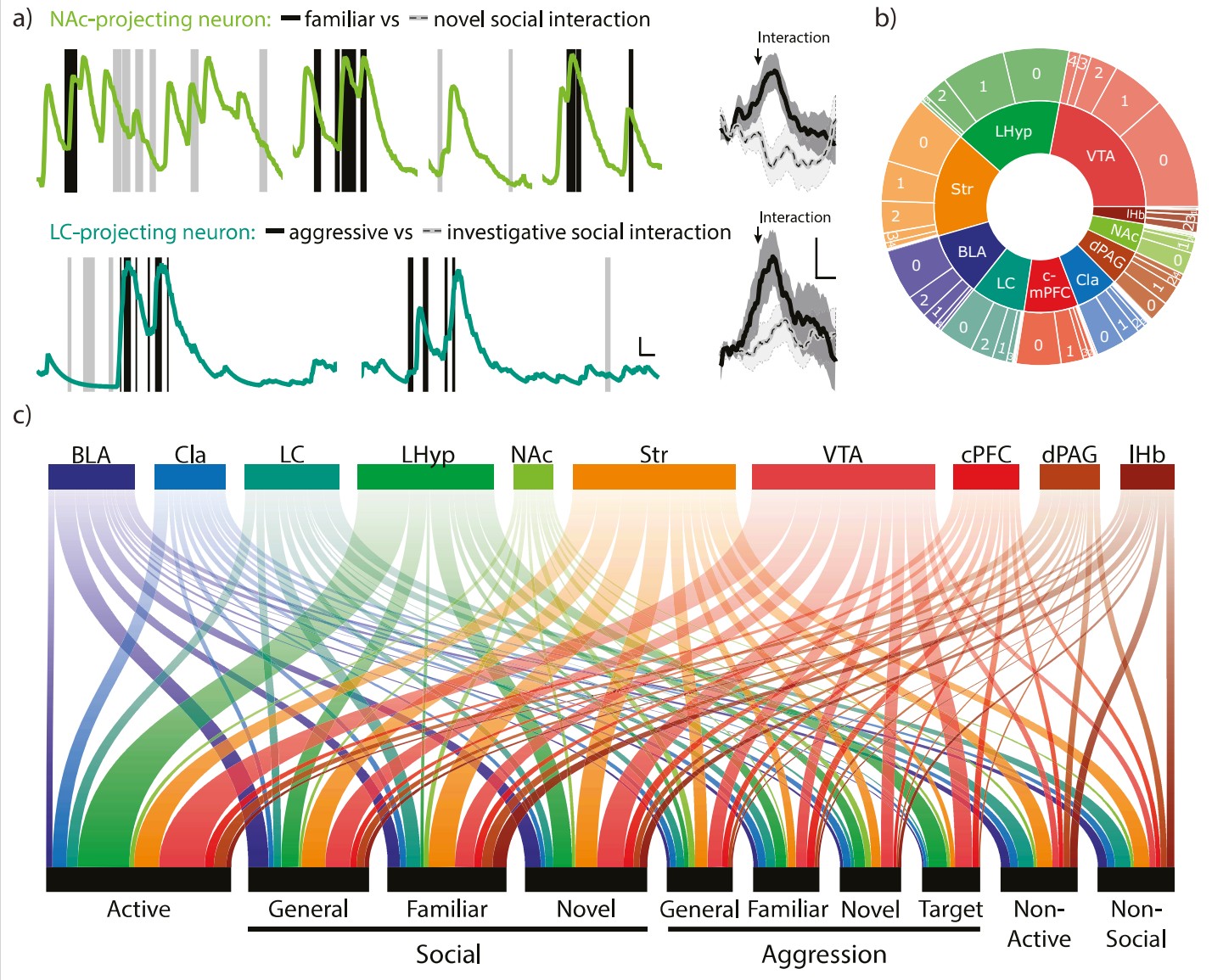

**Figure 5.** Neuronal cell-types vary in behavioral encoding. (**a**) Representative trace (left) and averaged ΔF/F traces (right) of calcium transients time-locked to behavioral annotations. Top traces depict a nucleus accumbens-projecting neuron with annotations denoting social interaction with either a familiar (black) or novel (gray) conspecific. Bottom traces denote a locus coeruleus-projecting neuron with annotations denoting aggressive (black) or investigative (gray) social interactions, regardless of conspecific target. Scale bars = 10 ΔF/F y-axis, 2 s (x-axis, left), and 5 ΔF/F (y-axis), 1 s (x-axis, right). N=34 behavioral epochs for familiar, 38 epochs for novel, 32 epochs for aggressive, and 53 epochs for investigative interactions. Mean ± SEM. Two-sided t-test: familiar max response vs. novel, $p$=0.0011; aggressive vs. investigative, $p$=0.049. (**b**) Distribution of the number of different behaviors for which each neuronal cell type statistically modifies its firing rate. (**c**) Schematic of neuronal cell-types and the behavioral categories for which each cell encodes.

The online version of this article includes the following figure supplement(s) for figure 5:

**Figure supplement 1.** Detailed view of neural cell types and behavioral encoding.

versus investigative social interactions (*Figure 5a*). Furthermore, certain populations display greater selectivity for what behaviors they encode compared to others, as shown by looking at the number of distinct behaviors for which individual neurons statistically modulated their calcium activity. Indeed, lHyp-projecting neurons are either not selective or selective for only one specific type of behavior, whereas some Str-projecting neurons significantly modulate their activity for up to six unique behaviors (*Figure 5b and c*). While examples of selective neurons of each type could be found for most behaviors, encoding was enhanced in certain populations. Using this approach, our results indicate

that Str-projecting neurons are preferentially modulated for aggressive behaviors, most notably with familiar conspecifics (*Figure 5c*, *Figure 5—figure supplement 1b*) as they statistically decreased their firing rate during the behavior (*Figure 5—figure supplement 1b*).

## Performance under reduced fluorophore complexity

To evaluate the performance of Neuroplex under more typical experimental regimes with reduced complexity, we applied the pipeline to two GCaMP transgenic animals injected with a subset of four fluorophores: mTagBFP2 (c-mPFC), mVenus (Str), mOrange2 (Cla), and mNeptune2.5 (VTA) (*Figure 6a*). Detection frequency and identification accuracy were assessed using both experimental and modeled datasets.

Despite differences in the number of functionally defined neurons between animals, Neuroplex successfully detected all four injected fluorophores in both subjects, with expected variability in labeling efficiency across regions (*Figure 6b and c*). Across both animals, 57% of functionally defined neurons were assigned a fluorophore identity, and 15% of these were classified as co-expressing a second fluorophore - consistent with expectations based on known projection overlap in the mPFC (*Figure 6d and f*).

To estimate classification accuracy, we generated model datasets that approximated experimental conditions by replicating observed fluorophore distributions, background fluorescence, GCaMP co-expression and added Gaussian white noise. When restricted to a single fluorophore identification, the pipeline achieved approximately 90% accuracy with fewer than 10% false positives (*Figure 6g*). When applied to simulated dual-labeled cells, Neuroplex correctly identified at least one fluorophore in 92% of cases and successfully resolved both fluorophores in 20% (*Figure 6h*).

Finally, we assessed how specific fluorophore pairings influenced co-identification performance. As expected, spectrally distinct fluorophores (such as mTagBFP2 and mNeptune2.5) were more reliably separated than spectrally overlapping pairs (such as mVenus and mOrange2) (*Figure 6i*). These results validate Neuroplex's ability to generalize to reduced-complexity labeling strategies and emphasize the importance of strategic fluorophore selection based on co-expression likelihood and system-specific constraints.

## Discussion

Our study establishes a framework for high-dimensional cell-type-resolved functional imaging in freely behaving animals. By integrating miniscope recordings with multiplexed confocal spectral finger-printing through the same GRIN lens, Neuroplex addresses key spectral limitations of head-mounted microscopy—enabling simultaneous distinction of nine neuronal populations alongside GCaMP activity. We demonstrate that Neuroplex allows for the identification of up to nine fluorophore-labeled neuronal subtypes during in vivo calcium imaging through a GRIN lens (*Video 1*). By combining functional recordings from a head-mounted miniscope with multiplexed spectral confocal imaging in the same animal, Neuroplex supports the assignment of fluorophore identity to functionally defined neuronal locations without relying on post-fixation tissue processing. This approach allows for simultaneous monitoring of diverse neuronal populations within a single subject, facilitating direct comparisons of activity patterns across cell types during behavior. In this study, we demonstrate classification of up to ten spectrally distinct fluorophores—including the nine injected fluorophores and genetically-expressed GCaMP—in medial prefrontal cortex (mPFC) neurons defined by their downstream projection targets.

While several existing methods identify neuronal subtypes following behavior, most rely on post hoc approaches, such as immunohistochemistry or in situ labeling. These techniques require co-registration between live imaging and fixed tissue, a process complicated by chromatic aberrations from the GRIN lens and nonuniform tissue distortion during fixation and mounting. In practice, this often necessitates labor-intensive manual, nonlinear alignment. Neuroplex circumvents these limitations by acquiring both structural and functional data in vivo through the same GRIN lens, using matched fields of view and consistent optical paths. Image registration is performed using a Python-based workflow that automatically computes optimal linear transformations—guided by anatomical landmarks, such as blood vessel patterns—ensuring reproducible alignment across animals and sessions. This approach requires only x/y shifts, rotation, and scaling, avoiding the need for nonlinear warping or manual

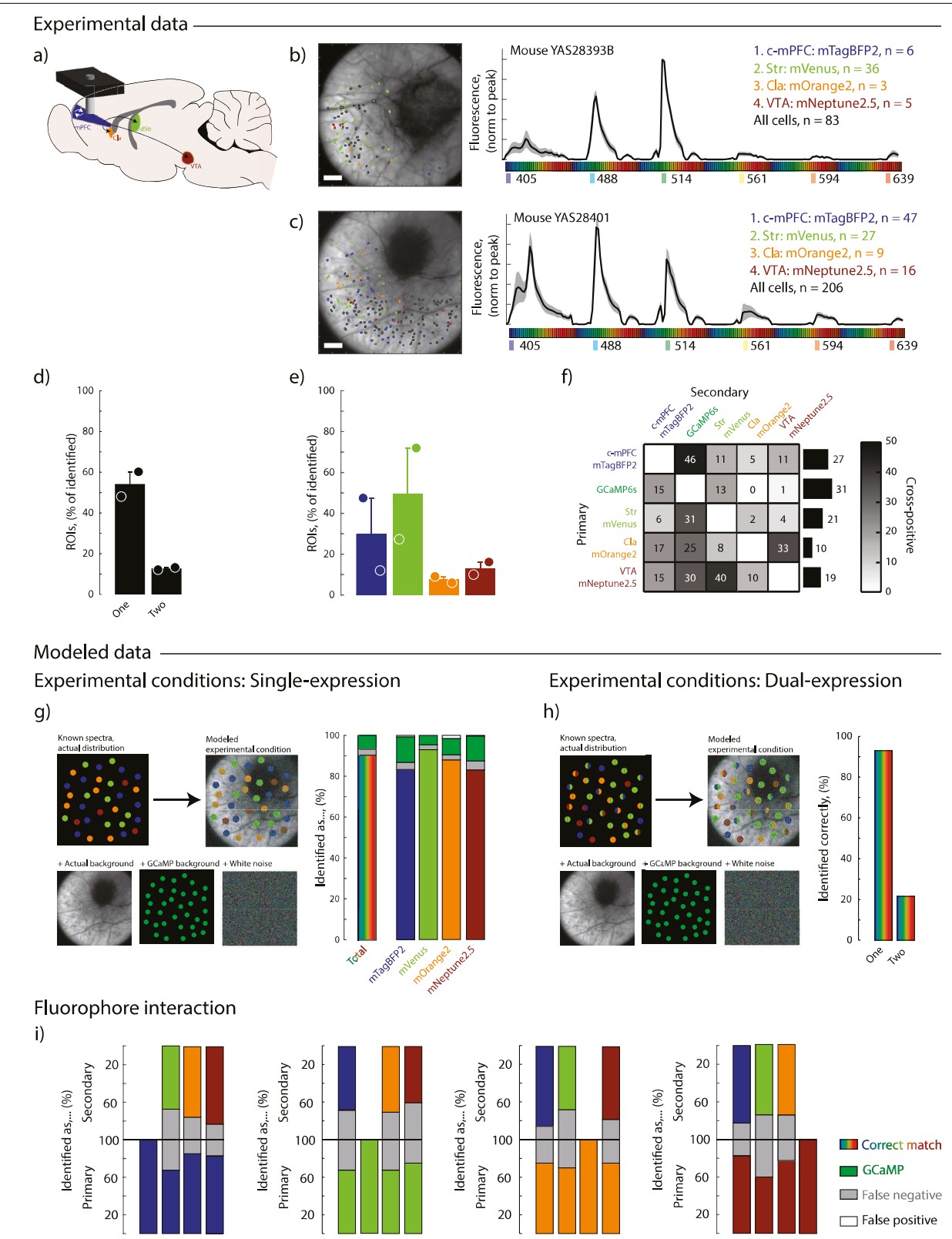

**Figure 6.** Performance of Neuroplex under reduced fluorophore complexity. (**a**) Schematic of experimental design: Four retrograde fluorophores were injected into distinct brain regions of GCaMP6s transgenic mice to label projection-defined pyramidal neurons in the medial prefrontal cortex (mPFC). mTagBFP2 was injected into the contralateral prefrontal cortex (c-mPFC), mVenus into the striatum (Str), mOrange2 into the claustrum (Cla), and mNeptune2.5 into the ventral tegmental area (VTA). (**b–c**) Spatial distribution of identified regions of interests (ROIs) overlaid on anatomical reference

*Figure 6 continued*

images (left) and corresponding average spectral fingerprints from each experimental animal (right). Shaded regions represent ±1.5 std from the mean. Scale bars 100 μm. (**d**) Proportion of ROIs classified as single- or dual-labeled based on dual-pass thresholding. Animal n=2; ROI n=289. (**e**) Percent of identified ROIs assigned to each individual fluorophore. (**f**) Frequency of dual-fluorophore assignments across the dataset. Left: Heatmap showing pairwise co-occurrence rates between fluorophore (and region) combinations. Right: Total frequency of dual hits per individual fluorophore/region. (**g–h**) Modeled experimental conditions assessing classification accuracy in either single-fluorophore (**g**) or dual-fluorophore (**h**) expression contexts. Spectra were simulated with empirical fluorophore distributions, real background, GCaMP contamination, and Gaussian white noise (left). Breakdown of error types (right). ROIs n=460, replicates n=100. (**i**) Fluorophore interaction analysis showing classification accuracy when fluorophores were co-expressed. Bars indicate the percentage of ROIs correctly identified (colored), missed (gray: false negatives), or incorrectly labeled (white: false positives).

intervention if care is taken to ensure parallel alignment between the GRIN surface and the detector. By eliminating post-fixation alignment, Neuroplex not only improves accuracy but also preserves animals for longitudinal studies, greatly enhancing experimental efficiency and reproducibility.

A key advantage of Neuroplex is its compatibility with longitudinal designs. Because both functional and spectral imaging are performed in vivo through the same implanted GRIN lens, the same neuronal populations can be assessed across multiple time points within a single subject. This enables chronic tracking of fluorophore-defined subtypes as animals undergo learning, disease progression, or therapeutic intervention—offering a level of circuit resolution not feasible with post-fixation methods. Additionally, spectral imaging can be performed prior to behavioral testing, allowing researchers to verify expression patterns and exclude poorly labeled subjects before committing to lengthy experiments. One important consideration is that multiplexed spectral imaging requires animals to be anesthetized, particularly when using fluorophores that spectrally overlap with GCaMP. Without silencing, large fluorescence transients from GCaMP can obscure dimmer fluorophore signals and reduce unmixing fidelity, leading to an increased false positive rate. Fluorophore identity in this framework is genetically encoded via retrograde AAV delivery and is, therefore, expected to remain stable across behavioral and spectral imaging sessions. Because both functional and spectral data are acquired in vivo through the same GRIN lens and co-registered using anatomical landmarks, assignment stability is not expected to vary across time unless expression levels change substantially. While repeat spectral imaging was not performed as a formal longitudinal experiment in this study, the stability of fluorescent protein expression supports the assumption that fluorophore identity reflects a persistent cellular attribute.

Neuroplex is compatible with a variety of GRIN lens types and sizes, as well as with cranial windows. In this study, we evaluated several commonly used GRIN lenses and found that all were usable when proper chromatic corrections were applied. We selected the widely used 1×4 mm silver-doped GRIN lenses for our experiments, given their high numerical aperture (NA) and reliable performance in vivo. While lithium-doped lenses exhibited improved spectral characteristics, including reduced chromatic aberrations, reduced wavelength-dependent transmission, and lower background fluorescence, they also had significantly lower NA, resulting in substantially dimmer signals. This tradeoff rendered them impractical for many in vivo settings, where fluorophore brightness and signal-to-noise ratio are often less than ideal. In practice, any GRIN lens that provides adequate optical throughput and allows for full-spectrum z-stacks within the microscope's travel range can be used with Neuroplex, provided appropriate chromatic correction is applied.

The current classification framework relies on linear unmixing followed by empirically defined thresholding rather than full probabilistic inference. This approach provides transparency and practical robustness under realistic noise and background conditions but does not generate single-ROI posterior uncertainty estimates. While formal conditioning metrics were not explicitly computed, empirical fingerprint acquisition and simulation-based perturbation analyses demonstrate sufficient spectral independence for reliable linear unmixing under the tested regimes.

Because the present study is designed to establish methodological feasibility rather than precise anatomical quantification, simulation-derived false-positive and false-negative regimes were not formally propagated into confidence bounds on subtype proportions or behavioral effect sizes. Accordingly, subtype fractions should be interpreted as assignment-dependent estimates rather than definitive anatomical measurements. Future implementations could incorporate Bayesian or likelihood-based classifiers to generate posterior identity probabilities and enable formal uncertainty propagation when quantitative estimation of projection convergence is central to the biological question.

Neuroplex is scalable and broadly adaptable to diverse circuit-mapping goals. While this study focused on projection-defined pyramidal neurons in the mPFC using nine fluorophores plus GCaMP, the pipeline is equally compatible with genetically defined subtypes, developmental stages, or activity-tagged populations. Many experiments may require fewer labels, allowing users to prioritize only the brightest and most spectrally distinct fluorophores for enhanced separation fidelity. This can be especially advantageous in experiments where multiple labels are expected within a single neuron—such as studies of projection convergence or co-expression—where careful fluorophore selection can substantially improve identification accuracy. In two experimental animals expressing four fluorophores plus GCaMP, Neuroplex maintained high classification accuracy (~90%) with low false positive rates and successfully identified dual-labeled neurons, demonstrating strong performance even under reduced spectral complexity. These results highlight the flexibility of the approach for both high- and low-complexity applications, across a variety of biological contexts and imaging constraints.

Dual-label assignments are most reliable when fluorophores are spectrally well separated and when signal-to-noise ratios are high. In contrast, spectrally adjacent fluorophore pairs or densely labeled regimes increase ambiguity and false-positive risk. Experimental design should, therefore, prioritize pairing spectrally distant fluorophores when projection convergence is of primary interest. Fluorophore selection can also be tailored to the excitation and emission capabilities of individual imaging systems. For instance, microscopes equipped with near-infrared detection may benefit from using fluorophores, such as iFP2.0, which emit more efficiently in the 700–740 nm range compared to mNeptune2.5.

Spectral unmixing operates on CNMF-derived ROI masks treated as fixed supports. Accordingly, segmentation quality, neuropil contamination, and partial overlap between neighboring cells can influence extracted spectral fingerprints and may contribute to false negatives or secondary assignments, particularly in densely labeled regions. These structured sources of uncertainty are expected to have the greatest impact under regimes of extreme class imbalance, low fluorophore brightness, strong neuropil signal, or pairing of spectrally overlapping reporters. Use of refined segmentation strategies or nuclear-localized reporters could reduce such structured uncertainty in future implementations.

More broadly, Neuroplex is expected to perform most robustly in regimes characterized by moderate projection convergence, balanced fluorophore representation, bright and spectrally distinct reporters, and adequate signal-to-noise ratio. Imaging directly within a projection target that has received dense retrograde labeling may introduce substantial class imbalance, which simulations predict will reduce detection sensitivity for the dominant fluorophore. In such cases, conservative assignment strategies, reduced spectral complexity, or refinement of ROI definition may improve interpretability. Careful fluorophore selection and pilot validation under intended imaging conditions are, therefore, recommended prior to large-scale application. Future implementations incorporating nuclear-localized reporters may further reduce segmentation-dependent ambiguity by constraining spectral signals to somatic compartments.

Relative fluorophore representation within the imaged field of view influences classification robustness. As demonstrated in our simulations of class imbalance (*Figure 3—figure supplement 3*), extreme over-representation of a single fluorophore primarily increases false-negative rates due to baseline normalization effects. In the present study, we intentionally avoided imaging directly within heavily infected projection targets (e.g. contralateral mPFC) in order to maintain moderate fluorophore representation across ROIs. Imaging in a densely labeled region would represent a more challenging regime, and we would expect reduced sensitivity for the dominant fluorophore under such conditions.

Cortical pyramidal neurons frequently collateralize to multiple downstream targets, and accordingly some ROIs exceeded the threshold for more than one fluorophore. In this proof-of-principle implementation, we adopted a specificity-first winner-take-all assignment rule for primary analyses to minimize false-positive multi-label calls under realistic noise conditions. This strategy likely underestimates the true prevalence of dual-projecting neurons and should, therefore, be interpreted as a conservative stratification approach rather than a statement of projection exclusivity. Accurate quantification of projection convergence would benefit from orthogonal ground-truth validation (e.g. paired tracers or staged injections) to establish confusion matrices for dual positives and to calibrate thresholds or priors. Multi-hit events can reflect true biological collateralization but may also arise from structured sources of ambiguity, such as neuropil contamination, partial ROI overlap, or imperfect

ROI boundaries. These factors may bias spectral estimates and contribute to secondary assignments, particularly in densely labeled regions. Practical mitigation strategies include conservative assignment rules, improved segmentation, and use of nuclear-localized reporters to reduce neuropil contribution.

Neuroplex is not designed for simultaneous multi-color functional imaging (e.g. combinations of spectrally distinct GECIs or GEVIs) during miniscope recordings. In head-mounted systems, reliable spectral separation is limited by restricted detection bandwidth, lack of multiplexed acquisition, and photon constraints, which prevent accurate unmixing of overlapping signals. Chromatic aberrations from GRIN lenses would also require wavelength-dependent focal adjustments incompatible with dynamic in vivo imaging. While spectral confocal systems can resolve fluorophore identity, they require head fixation and lack the temporal resolution needed to capture fast neural dynamics in freely behaving animals. Moreover, applying spectral unmixing to time-varying signals would be unstable, as dynamic fluorescence violates the assumptions of linear unmixing. Neuroplex, therefore, decouples functional recording from spectral identity assignment, enabling robust activity measurement alongside high-dimensional post hoc cell-type classification.

The ability to co-register multiple fluorophore markers with functional calcium data will substantially enhance not only the throughput of neural activity studies, but also the statistical power of circuit-level analyses. By enabling multiplexed classification of neuronal subtypes within the same animal, Neuroplex facilitates more precise, efficient, and scalable interrogation of brain function. Furthermore, the pipeline's compatibility with longitudinal studies and avoidance of post-hoc tissue processing make it particularly valuable for investigating dynamic processes like learning, adaptation, or disease progression at circuit-level resolution. Overall, our integrative approach opens new avenues for dissecting the cellular architecture of behavior across time, space, and experimental contexts.

# Materials and methods

**Key resources table**

| Reagent type (species) or resource | Designation | Source or reference | Identifiers | Additional information |
|---|---|---|---|---|
| Strain, strain background (*Mus musculus*) | B6.DBA-Tg(tetO-GCaMP6s)2Niell/J mice | Jackson Labs | RRID:IMSR_JAX:024742 | |
| Strain, strain background (*Mus musculus*) | B6.Cg-Tg(Camk2a-tTA)1Mmay/DboJ mice | Jackson Labs | RRID:IMSR_JAX:007004 | |
| cell line (Homo- sapiens) | HEK 293T | GE Dharmacon, Fisher Scientific | NC0260915 | |
| Software, algorithm | Inscopix Data Processing Software (IDPS) | Bruker | | Miniscope analysis software |
| Recombinant DNA reagent (Sapphire) | mT-Sapphire-C1 | Addgene | RRID:Addgene_54545 | |
| Recombinant DNA reagent (Orange2) | mOrange2-N1 | Addgene | RRID:Addgene_54568 | |
| Recombinant DNA reagent (mNeptune2.5) | pcDNA3-mNeptune2.5 | Addgene | RRID:Addgene_51310 | |
| Recombinant DNA reagent (Scarlet) | pmScarlet-H_C1 | Addgene | RRID:Addgene_85043 | |
| Recombinant DNA reagent, (mtagbfp2) | pCAG-mTagBFP2 | Addgene | RRID:Addgene_122373 | |
| Recombinant DNA reagent, (mVenus) | mVenus | Addgene | RRID:Addgene_198192 | |
| Recombinant DNA reagent, (FusionRed) | FusionRed-pBAD | Addgene | RRID:Addgene_54677 | |
| Recombinant DNA reagent, (mCyRFP2) | CMV-mCyRFP2-CREB | Addgene | RRID:Addgene_137003 | |
| Recombinant DNA reagent, (mTurquoise2) | mTurquoise2 | Addgene | RRID:Addgene_198196 | |

*Continued on next page*

*Continued*

| Reagent type (species) or resource | Designation | Source or reference | Identifiers | Additional information |
|---|---|---|---|---|
| Other | ProView Integrated Lens 1x4 mm | Bruker | Bruker Part No: 1050–004637 | GRIN lens |
| Other | ProView Integrated Lens 0.6x7.3 mm | Bruker | Bruker Part No: 1050–004413 | GRIN lens |
| Other | ProView DC Integrated Lens 0.66x7.5 mm | Bruker | Bruker Part No: 1050–005442 | GRIN lens |
| Software, algorithm | Matlab | Mathworks | RRID:SCR_001622 | |
| Software, algorithm | Motr | https://github.com/motr/motr; *Ohayon et al., 2018*; *Ohayon et al., 2013* | | Mouse tracker |
| Software, algorithm | JAABA | https://github.com/kristinbranson/JAABA | Janelia Automatic Animal Behavior Annotator (RRID:SCR_027430) | Behavior annotator |
| Other | nVoke2 | Bruker | nVoke System (RRID:SCR_023028) | Miniscope platform |
| Software, algorithm | Neuroplex | https://github.com/Neurocipher/PythonPipeline | This study | |

## Code and data availability

Raw and processed data: https://bit.ly/NeuroplexData.

Code and data utilized in Neuroplex multispectral detection: GitHub - Neurocipher/PythonPipeline: Python pipeline for Neurocipher (copy archived at *Neurocipher, 2025*).

Code and data utilized in calcium/behavior correlation: https://github.com/MetaCell/Zeiss-Data-Science (copy archived at *MetaCell, 2025*).

A detailed tutorial on these processes can be found via this repository: https://zeiss.tourial.com/dc/MultiColorInVivoImaging.

## Ethics & inclusion statement

We have carefully considered research contributions and authorship criteria during collaborations so as to promote greater equality in research.

## Mice

All experimental procedures were approved by the Max Planck Florida Institute for Neuroscience Institutional Animal Care and Use Committee (#24–002) and were performed in accordance with guidelines from the US NIH. Mice were group housed in 12 hr light-dark cycle with food and water ad libitum. The mice used in this study resulted from crossing heterozygous or homozygous B6.DBA-Tg(tetO-GCaMP6s)2Niell/J mice (JAX 024742) to heterozygous B6.Cg-Tg(Camk2a-tTA)1Mmay/DboJ (JAX 007004).

## Fluorophore selection

### Initial selection

Fluorophores were identified based on spectral profiles published on fpbase.com and chosen for further study based on the specific spectral fingerprints we hypothesized would be distinguishable by multiplexed spectral imaging. These fluorophores included: mTagBFP2, mTurquoise2, T-Sapphire, mVenus, mPapaya, mOrange2, mScarlet, FusionRed, mCyRFP1, and mNeptune2.5.

### In vitro expression

Vectors containing the identified fluorophores obtained from Addgene (Addgene.com) were cloned into plasmids under the CMV promoter. HEK293T cells (GE Dharmacon, Fisher Scientific) were cultured in DMEM supplemented with 10% FBS at 37 °C in 5% $CO_2$ and transfected with plasmids using Lipofectamine 1000 (Invitrogen). Imaging was performed 24–48 hr following transfection. HEK293T cells

**Table 1.** AAV<sub>retro</sub> fluorophore evaluation.

| Fluorophore | Laser | Interfering Fluorophore | Reason for ranking |
|---|---|---|---|
| mOrange2 | 514 nm | NA | Bright, distinct |
| mTagBFP2 | 405 nm | NA | Several distinct spectral bins |
| mVenus | 514 nm | NA | Bright |
| mTurquoise2 | 405 nm *Would benefit from 455 nm | mTagBFP2 & T-Sapphire | Neutral |
| T-Sapphire | 405 nm | NA | Neutral |
| FusionRed | 594 nm | mScarlet | Underperformer, slightly dim |
| mScarlet | 561 nm | mOrange2 & mNeptune2.5 | Cross talk |
| mNeptune2.5 | 639 nm | mScarlet | Dim, cross talk |
| mCyRFP2 | 488 nm | NA | Very dim |

were used as an expression platform only and were not rigorously tested for potential contamination from other cell lines.

## Viral construction and evaluation

Plasmids encoding mTagBFP2, mTurquoise2, T-Sapphire, mPapaya, mOrange2, mScarlet, FusionRed, mCyRFP1, and mNeptune2.5 were introduced into a hSyn-mVenus vector, replacing the mVenus sequence. The resulting vectors were used to create AAV<sub>retros</sub> (UNC vector core). Each virus was tested by injecting it into the right medial prefrontal cortex of a wild-type mouse. After waiting three weeks for expression, mice were euthanized and 100 μm coronal sections containing the medial prefrontal cortex were prepared. Fluorophores were evaluated for expression levels, brightness, photostability, and neuronal death. mPapaya was excluded from further experiments due to excessive neuronal death after viral expression.

## Surgery

Mice were anesthetized with 4% isoflurane vapor in 100% oxygen gas and maintained with 1–2.5% isoflurane vapor in 100% oxygen gas mixtures. Mice were aligned in a stereotactic frame (Kopf Instruments), and their body temperature was measured with a rectal probe and maintained at 37°C with a heating pad. Warmed sterile saline was injected subcutaneously at a rate of 0.1 mL/hr to maintain hydration. Additionally, mice received subcutaneous injections of carprofen (5 mg/kg) and dexamethasone (0.2 mg/kg) to reduce inflammation. Mice were monitored post-surgically and returned to their respective homecage once ambulation returned to normal.

### For viral evaluation

A midline incision was made down the scalp, and a dental drill was used to perform a small craniotomy over the right medial prefrontal cortex. A 2.5 μL syringe (Hamilton Company) was used to inject 250 nL of virus at a rate of 0.25 μL/min using a microsyringe pump (UMP3 UltraMicroPump, Micro4; World Precision Instruments), for coordinates see *Table 1*. The needle was slowly extracted from the injection site over 10 min. The scalp was closed using surgical glue.

### For multi-color GRIN experiments

A midline incision was made down the scalp, the scalp lightly scored to improve adherence of dental cement, and a dental drill used to perform a small craniotomy over the target area. A 2.5 μL syringe (Hamilton Company) was used to inject viruses at a rate of 0.25 μL/min using a microsyringe pump (UMP3 UltraMicroPump, Micro4; World Precision Instruments), see *Table 2* for coordinates and volumes. The needle was slowly extracted from the injection site over 10 min. This injection procedure was repeated for each of the nine fluorophore viruses, each injected into a unique region. Once all

**Table 2.** Injection coordinates.

| Acronym | Region | Angle | Medial/ Lateral | Anterior/ Posterior | Dorsal/ Ventral | Volume |
|---|---|---|---|---|---|---|
| dPAG | Dorsal periaqueductal gray | 26° | 1.18 | –4.2 | 2.36 | 300 nL |
| BLA | Basolateral amygdala | 0° | 2.95 | –1.6 | 3.8<br>3.3 | 250 nL<br>250 nL |
| NAc | Nucleus accumbens | 0° | 1.0 | 1.3 | 4.0<br>3.5 | 250 nL<br>250 nL |
| Str | Striatum | 0° | 1.25 | 1.3 | 2.5<br>2.0 | 250 nL<br>250 nL |
| Cla | Claustrum | 0° | 2.2 | –1.7 | 2.6 | 300 nL |
| LHyp | Lateral hypothalamus | 0° | 1.1 | –1.3 | 5.3<br>4.75 | 250 nL<br>250 nL |
| LHb | Lateral habenula | 0° | 0.5 | –1.6 | 2.75<br>2.5 | 250 nL<br>250 nL |
| LC | Locus coeruleus | 0° | 0.8 | –5.3 | 3.0<br>3.0 | 250 nL<br>250 nL |
| VTA | Ventral tegmental area | 0° | 0.45 | –3.0 | 4.25<br>3.45 | 250 nL<br>250 nL |
| c-mPFC | Contralateral medial prefrontal cortex | 0° | 0.4 | 1.5 | 1.45 | 300 nL |

viral injections were complete, a 1.2 mm diameter craniotomy was performed over the mPFC and the dura was carefully removed. A custom-made metal probe measuring 1 mm in diameter was lowered at a rate of 100 μm/min to reach 300 μm above the desired imaging plane. The probe was left in place for 15 min before being retracted slowly over 10 min. Immediately afterward, a 1×4 mm silver-doped GRIN lens with integrated baseplate ('1×4 mm regular,' Inscopix) and head-bar was lowered into place at a rate of 100 μm/min. Any area between the craniotomy and GRIN lens was sealed with silicone (KWIK-SIL, World Precision Instruments) after which the implant was secured in place using dental cement. The skin was sutured around the implant (DemeTech). Mice were given at least 5 weeks to recover from surgery before use in experiments.

## Mouse perfusion

Mice were anesthetized with an intraperitoneal (i.p.) injection of a ketamine (100 mg/kg) and xylazine (50 mg/kg) and transcardially perfused with ice-cold 1 X phosphate-buffered saline (PBS), followed by ice-cold 4% paraformaldehyde (PFA) in 1 X PBS. The brain was dissected and post-fixed in 4% PFA overnight at 4 °C. Brains were sectioned at 100 μm with a vibratome (VT1200, Leica) and mounted with Vectashield mounting media (Vector Laboratories).

## Behavioral experiments

### Testing

One week prior to testing, the back of sentinel mice was dyed with blonde hair dye (Born Blonde Maxi, Clairol) with differing patterns for tracking by computer vision. The behavioral paradigm consisted of 5 days, with one behavior session per day. At the beginning of each behavioral session, test mice were acclimated to the behavior chamber alone for 10 min. On days 1–4, the same sentinel mouse (initially novel) was added to the behavior chamber following the acclimation period and allowed to freely interact for 10 min. On day 5, the trained sentinel mouse and a second novel sentinel mouse were added to the behavior chamber following the acclimation period and allowed to freely interact for 10 min. The test box was cleaned and filled with new bedding between each session. Each sentinel mouse interacted with a maximum of three test mice per 5 day paradigm. Custom-written MATLAB (Mathworks) code was used to record behavioral videos at 20 Hz.

## Analysis

After all mice had been tested, sentinel mice were individually videotaped for 10 min for generating training data. Individual and test videos were fed to the *Motr* program (https://github.com/motr/motr; *Ohayon et al., 2013*) to create tracks that were sent to *JAABA* (https://github.com/kristinbranson/JAABA; *Kabra et al., 2013*) Janelia Automatic Animal Behavior Annotator (RRID:SCR_027430) for unbiased computer identification of behaviors. *JAABA* classifiers were trained on pilot data sets.

## Imaging

### In vivo calcium imaging using the miniscope

On each day of behavioral experimentation, the miniscope was mounted on the implanted GRIN lens and baseplate immediately prior to the mouse being placed in the behavioral chamber. Custom-written MATLAB code triggered simultaneous video acquisition and nVoke2 (Inscopix) calcium recording, both recording at 20 Hz sampling rate. Parameters of the miniscope, such as the LED power, the gain, and the electronic focus, were adjusted on a mouse-to-mouse basis but otherwise kept consistent for the five sequential days of behavioral testing.

Acquired calcium transients, concatenated per recording day with each recording day consisting of the acclimation and behavior time, were processed in the Inscopix Data Processing Software (IDPS, Inscopix). First, the traces were spatially down sampled by a factor of 2, then preprocessed, spatially filtered, and motion corrected. Individual cells were identified using a constrained non-negative matrix factorization algorithm (CNMF). Traces with abnormal physiological calcium transients (i.e. transients lasting over 1 min) were excluded. For co-registration purposes, the motion-corrected video was temporally averaged into an image depicting anatomical landmarks. ROIs generated by the CNMF and their corresponding calcium traces were also exported.

### Multicolor volumetric confocal imaging

The post-behavioral confocal imaging was performed using a LSM 980 confocal microscope on an Examiner Z.1 upright stage. The mouse treadmill was inserted directly onto the xy-mechanical stage, bypassing the z-piezo stage. Images were taken using a 10 x, NA 0.4 objective lens (C Epiplan-Apochromat, 422642–9900, Zeiss), and utilizing both multialkali PMTs (Ch1 and Ch2) and the GaAsP detector (ChS), spanning a wavelength range from 350–750 nm resulting in 34 spectral bins. All six excitation lasers (405, 488, 514, 561, 594, and 639 nm) were used during volumetric spectral imaging. The pinhole was set to 'optimal' when imaging without a GRIN lens, and to 350 µm when imaging through a GRIN lens. Detector gain voltages and laser powers were set for each animal to prevent detector saturation for the brightest spectral channels and kept constant for each multiplexed spectral image within each animal. MultiBeam splitters (MBS) were set to MBS 405, MBS 488/561/639, or MBS 455/514/594 depending on the excitation laser used. No dichroic secondary beam splitter was used. Imaging parameters included a zoom setting of 1.0, no averaging, and a scan speed setting of 5.

### GRIN lens optical transmission determination

A custom-built GRIN lens micromanipulator was used to suspend the GRIN lens above a platform. Using a photodiode power sensor (S121C, Thorlabs) and power meter (PM100D, Thorlabs), we recorded the laser power of the LSM 980 excitation lasers (ZEISS) out of the objective lens, first directly and second after focusing through a GRIN lens. The difference in these values was used to calculate the wavelength-dependent transmission through the GRIN lens. The data was fitted to a second-order polynomial and subsequently used to pre-adjust the intensity of each excitation laser to ensure wavelength-independent laser power on the sample.

### Measuring GRIN-induced chromatic aberrations

Using a custom-built GRIN lens micromanipulator (MPFI), the GRIN lens was centered and suspended over the fluorescent 'field of rings' pattern on an Argo-LM v2.0 slide (ArgoLight). The pattern was imaged through the GRIN lens using a 10 x objective lens (0.4 NA, C Epiplan-Apochromat, ZEISS). The upper and lower z-stack limits were set using the 405 nm and 639 nm lasers, respectively, after which the entire z-stack was imaged at 5 µm intervals for each excitation laser (405, 488, 514, 561, 594, and 639 nm). To determine the chromatic shifts along the optical axis, the z-plane with the brightest

intensity for the center crosshair pattern was determined to be the focal plane for each laser wavelength, and the wavelength – z focal plane relationship was fit to these values using a second-order polynomial. There were negligible chromatic aberrations in the optical plane (i.e. the xy dimensions).

## Multiplexed spectral imaging technique

To enable discrimination of all 10 fluorophores, we utilized a 'multiplexed spectral imaging' technique. This involved sequentially imaging the same field of view using six different excitation lasers (405, 488, 514, 561, 594, and 639 nm) while detecting the emission in spectral mode using both multialkali (32 array detector) and the GaAsP PMTs (two detectors), together collecting the fluorescence emissions in 34 separate spectral bins. The bin widths were approximately 10 nm between 400–695 nm (longer wavelengths have larger emission bins) using the GaAsP detector, with the multialkali PMTs spanning the wavelengths from 350–400 nm and 695–750 nm, respectively. This resulted in a spectral fingerprint consisting of 204 measurements per ROI (6 excitation lasers×34 emission bins).

## Spectral fingerprint generation

After allowing 24 hr for expression, wells containing fluorophore-expressing HEK293T cells were rinsed and filled with imaging buffer. Wells were imaged on a ZEISS LSM 980 at 10 x magnification using the multiplexed spectral technique. Laser powers were determined per sample but kept the same for each excitation laser. The pinhole was set to optimal and one focal plane and field of view were taken per sample.

For each fluorophore, the 6 corresponding images were imported into Fiji (ImageJ). The image with the highest intensity was selected and then thresholded to isolate cells from background. The Analyze Particles module was used to determine ROIs. The 204-point spectral fingerprint of each fluorophore was determined by averaging the multiplexed spectral traces from all ROIs from a single sample.

## In vivo multiplexed spectral imaging

Mice were anesthetized with an i.p. injection of ketamine (100 mg/kg) and xylazine (10 mg/kg) to reduce high-intensity fluctuations of GCaMP6s transients, though low-intensity slow-wave fluctuations remained. The GRIN lens was thoroughly cleaned with isopropanol and water and the mouse was head-fixed on a custom treadmill (MPFI). The temperature was monitored and maintained with hand warmers (Hothands). Images were obtained using the spectral imaging mode on a ZEISS LSM 980 confocal microscope running the ZEN Blue software. Using 10 x magnification, the objective pinhole was set to 350 µm to minimize z-dimensional chromatic aberration of emission introduced by the GRIN lens. For each mouse, z-range was set by utilizing the 405 nm and 639 nm excitation lasers to account for the remaining z-dimensional chromatic aberration and sampled at 5 µm intervals with the typical range being 48 slices or 240 µm. The maximum intensity of all fluorophores was found, and the laser power of the corresponding laser was set to avoid oversaturation. Using this laser value, the remaining laser values were adjusted to account for differing wavelength-dependent transmission of the GRIN lens except for the 639 nm laser. Because mNeptune2.5 is the only fluorophore to emit following 639 nm excitation, and due to the reduced efficiency of its excitation at this wavelength, we consistently set the 639 nm laser to 40% power across all mice. Beginning with the 405 nm laser and progressing sequentially, we recorded the entire z-stack in the spectral imaging mode for each laser. If high-intensity fluctuations for GCaMP were observed, the mouse was recorded on a subsequent day to eliminate these as much as possible.

In post-processing, spectral z-stacks from each laser were cropped in the z-dimension to remove any high-noise z-planes, cropping was applied consistently to all images from the same mouse. Background subtraction was performed to reduce non-somatic signals. A summed Z-projection for each excitation laser stack was used for the spectral analysis.

## Co-registration of functionally defined neurons

### nVoke to Zeiss registration

Prior to image registration, a series of preprocessing steps were applied to denoise and enhance salient structural features. A Gaussian filter with a small sigma value was first applied to reduce noise,

with a sigma of 1 pixel applied to calcium imaging data and 2 pixels applied to confocal imaging data. To estimate and remove background, a Gaussian blur with a large sigma was applied to the image, and the resulting blurred background was subtracted from the original. The sigma used for this background estimation was 50 pixels for calcium imaging and 100 pixels for confocal imaging. Following background subtraction, a morphological black-hat operation was applied to extract dark, vessel-like features. This operation highlights structures that are smaller than a specified window size and is particularly effective at isolating blood vessels, which serve as reliable features for subsequent registration. A window size of 11 pixels was used for calcium imaging and 21 pixels for confocal imaging. Image registration was then formulated as an optimization problem designed to maximize correlation between images by adjusting four transformation parameters: x-translation, y-translation, rotation, and global scaling (i.e. a similarity transformation). The registration process was conducted in two stages. First, a coarse, exhaustive grid search explored a defined parameter space, with translation ranges set to ±60 pixels, rotation to ±15°, and scaling to a range of 1.8–2.0. All parameters were sampled at discrete intervals (5 pixels for translations, 5 degrees for rotation, and 0.05 for scaling). The best candidate from this global search was then refined using gradient descent to obtain a locally optimal solution. The learning rate for this fine-tuning stage was set to 0.5.

### Fluorophore identification

## Modeling of experimental variables to assess accuracy of algorithms

To assess how the fluorophore identification algorithm was performing, we utilized wells containing single fluorophore-expressing HEK293T cells imaged using the multispectral approach. We created a library of individual ROIs from each well and randomly sampled these ROIs to create simulated datasets. Each simulation evenly distributed the desired manipulation to ROIs of each fluorophore and was conducted in replicates of 100. Fluorophore matches assigned by the algorithm were compared to the known ROI identity, allowing us to measure the accuracy by determining the percentage of correct matches, false negatives (no match), and false positives (incorrect match). We tested the algorithm robustness under several simulated perturbations, including added background signal, unequal fluorophore distributions, and synthetic ROIs formed by combining spectral from different fluorophores. GCaMP background was modeled by adding scaled GCaMP spectra at varying intensities relative to the average fluorophore brightness, spectral background consisted of adding a uniform mixture of fluorophore spectra at varying proportions relative to the average fluorophore intensity, and Gaussian white noise was added at varying signal-to-noise ratios. Initially, we modeled each condition separately to determine which perturbations created what types of errors. Finally, we simulated an experimental condition by matching conditions within an animal subject as closely as possible. To this end, we recreated the fluorophore population of the dataset to reflect the actual measured hits, added GCaMP background at a level of 30% (the measured peak of GCaMP signal within ROIs), spectral background obtained by averaging all pixels within the GRIN lens FOV at a rate of 30% (measured by comparing the peak intensity of background to the average peak intensity of fluorophore-containing ROIs), and Gaussian white noise at a signal-to-noise ratio of six as an estimate.

To evaluate performance on dual-labeled ROIs, we generated datasets containing all pairwise combinations of two fluorophores. To this end, we randomly selected ROIs from HEK293T wells containing single fluorophores and created new ROIs by summing the spectra from two ROIs with each of the respective fluorophores. The resulting dataset contained equal numbers of ROIs expressing all possible dual-fluorophore pairs and ROIs expressing only a single fluorophore. The modeling framework was designed to characterize expected classification behavior across a range of experimental regimes, including background fluorescence, class imbalance, and reduced signal-to-noise ratio. These simulations provide practical performance guidance but were not used to compute formal error bars or propagate uncertainty into downstream biological analyses.

## Single pass algorithm

The multiplexed spectral data from the functionally defined ROIs was evaluated to fit to the pure sample fingerprints using a linear regression. Beta values for each fluorophore were generated for ROIs using both the raw emission values ($beta_{raw}$) and those normalized to maximum intensity ($beta_{norm}$). For each mouse, a cutoff was determined as 1.5 standard deviations above mean for each fluorophore's beta values. It was necessary to do this on a mouse-by-mouse basis because background varied based

on the number and ratio of neurons expressing fluorophores (*Figure 3—figure supplement 1*). ROIs with $beta_{raw}$ or $beta_{norm}$ above these thresholds were considered positive for a fluorophore. If multiple fluorophores exceeded the threshold for an ROI, the fluorophore with the largest z-scored beta value was assigned as the primary identity (winner-take-all rule). This conservative approach was chosen to prioritize specificity under realistic noise and background conditions. Additional above-threshold fluorophores were retained as 'secondary hits' but were not incorporated into primary subtype stratification analyses. Both $beta_{raw}$ and $beta_{norm}$ were used, as they exhibit complimentary biases for bright and dim ROIs, respectively. Cutoffs and procedures were evaluated using multiplexed spectral bins where only one fluorophore could be emitting.

### Dual pass algorithm

To recover fluorophore assignments missed by the first pass due to over-represented fluorophores, we implemented a second identification step for any ROIs which were not assigned a hit by the first pass. This second pass utilized the same linear regression and beta multiplier calculation as the first pass but adjusted these values to correct for uneven fluorophore distribution. To apply the correction, we first modeled a dataset with a perfect distribution of HEK293T cell ROIs from single fluorophore wells. We obtained the average beta multipliers for all ROIs and assessed the beta contribution of each fluorophore as a percent of total. These values were utilized as a theoretical equal distribution. We then determined the beta contribution for each fluorophore in our experimental conditions and adjusted the cutoff threshold for assignments based on the deviation from the theoretical value, resulting in proportionally lowered thresholds for over-represented fluorophores.

## Linking neural activity to behavior and delineating by cell-type

For each animal and session, we defined a behavior set appropriate to the session type. For example, single-animal behaviors (moving, being still) were defined for acclimation sessions, while social interaction behaviors (sniffing, aggression) were defined for training and testing sessions, with different conspecific targets (novel vs. familiar) in the testing session. Then, for each cell and each behavior in each session, a t-test was carried out comparing mean activity in the 2.5 s window before versus after behavioral onset with Bonferroni correction accounting for multiple comparisons across different behaviors. If the mean activity was significantly different before and after a given behavior, the cell was classified as behaviorally selective. The sunburst diagram visualizes the number of cells that were responsive to different behaviors, separated by different brain regions. The sankey diagram visualizes the relative proportion of cells that were responsive to different behaviors separated by different brain regions.

## Acknowledgements

We would like to acknowledge David Kloetzer for lab management, Yuki Hayano and Irena Suponitsky-Kroyter for technical assistance, and the MPFI ARC, including Elizabeth Garcia and Amanda Coldwell for animal care and maintenance. This work was supported by National Institutes of Health Grants R35-NS-116804 (RY) and F32MH120872 (MLP). Open access funding provided by Max Planck Society.

## Additional information

### Competing interests

Mary L Phillips: Mary Philips is affiliated with Zeiss Research Microscopy Solutions. The author has no other competing interests to declare. Zhe Dong: Zhe Dong is affiliated with Metacell. The author has no other competing interests to declare. The other authors declare that no competing interests exist.

## Funding

| Funder | Grant reference number | Author |
|---|---|---|
| National Institute of Neurological Disorders and Stroke | R35NS068410 | Ryohei Yasuda |
| National Institutes of Health | F32MH120872 | Mary L Phillips |

The funders had no role in study design, data collection and interpretation, or the decision to submit the work for publication.

## Author contributions

Mary L Phillips, Conceptualization, Resources, Data curation, Software, Formal analysis, Supervision, Funding acquisition, Validation, Investigation, Visualization, Methodology, Writing – original draft, Project administration, Writing – review and editing; Nicolai T Urban, Resources, Data curation, Formal analysis, Methodology, Project administration, Writing – review and editing; Taddeo Salemi, Formal analysis, Methodology; Zhe Dong, Software, Formal analysis, Validation, Visualization, Writing – review and editing; Ryohei Yasuda, Conceptualization, Supervision, Funding acquisition, Validation, Project administration, Writing – review and editing

### Author ORCIDs

Mary L Phillips ⬡ https://orcid.org/0000-0003-4696-1555
Nicolai T Urban ⬡ https://orcid.org/0000-0001-7643-9755
Ryohei Yasuda ⬡ https://orcid.org/0000-0001-6263-9297

### Ethics

All experimental procedures were approved by the Max Planck Florida Institute for Neuroscience Institutional Animal Care and Use Committee (#24-002) and were performed in accordance with guidelines from the US NIH.

Reviewer #1 (Public review): https://doi.org/10.7554/eLife.110277.3.sa1
Reviewer #2 (Public review): https://doi.org/10.7554/eLife.110277.3.sa2
Author response https://doi.org/10.7554/eLife.110277.3.sa3

# Additional files

### Supplementary files

MDAR checklist

### Data availability

All raw data are available at https://doi.org/10.5281/zenodo.17915226.

The following dataset was generated:

| Author(s) | Year | Dataset title | Dataset URL | Database and Identifier |
|---|---|---|---|---|
| Yasuda R, Phillips M, Urban N, Salemi T, Dong Z | 2025 | Dataset for: Functional imaging of nine distinct neuronal populations under a miniscope in freely behaving animals | https://doi.org/10.5281/zenodo.17915226 | Zenodo, 10.5281/zenodo.17915226 |

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
