## [Editor Report · eLife Assessment]

The new development of Neuroplex, a pipeline that links projection-defined neuronal identity to in vivo calcium activity within the same animal, is an **important** contribution to the field of neuroscience and beyond. The strength of evidence is **convincing**.

---

## [Referee Report · Reviewer #1 (Public review)]

Genetically encoded fluorescent proteins expressed in specific cell types allow recognising them in vivo and, if the protein is a functional indicator, as in the case of genetically encoded calcium indicators (GECIs), to record activity from the same cellular ensemble. Ideally, if proteins (fluorophores) have perfectly distinct spectral properties, signals can be distinguished from as many cell types as the number of employed fluorophores. In practice, fluorescent proteins have non-negligible crosstalk both in absorption and emission bands. In addition, fluorescence contribution of each fluorophore normally varies from cell to cell and therefore spectral properties of cells expressing two or more proteins are different. The work of Phillips et al. addresses this challenge. The authors present an approach defined as "Neuroplex", allowing identification of up to nine cell types from the same number of fluorophores. The fingerprint of each cell is then associated with functional fluorescence from the GECI GCaMP, allowing recording calcium activity from that specific cell. The method is implemented in vivo using head-mounted miniscopes.

The authors used a mouse line expressing GCaMP in cortical pyramidal neurons and developed an experimental pipeline. First, they injected the nine AAV viruses, causing expression of fluorophores in a different brain area. The idea was not to image that area, but a non-infected medial prefrontal cortex (mPFC) section where neurons could be infected by their axons projecting in an injected area, in this way being identified by their targeting region(s). A GRIN lens, allowing spectral analysis, was mounted in the mPFC section, and GCaMP fluorescence was then recorded during behavioural tasks and analysed to identify regions of interest (ROIs) corresponding to neuron somata. After functional imaging, the head of the mouse was fixed, spectral analysis was performed, and after necessary correction for chromatic distortions, the fluorophore contribution was determined for each ROI (neuron) from where GCaMP signals were detected. Notably, the procedures for estimation and correction of chromatic aberration and light transmission (described in Figure 2) were a major challenge in their technical achievements. The selection of the nine fluorophores was another big effort. This was done by combining computer simulations and direct measurement of spectra from individual proteins expressed in HEK293 cells. It is important to say that the authors could simulate arbitrary combinations of two or more different fluorophores and evaluate the ability of their algorithm to detect the correct proteins against wrong estimations of false-negative (absence of an expressed protein) or false-positive (presence of a non-expressed protein). Not surprisingly, this ability decreases with the level of GCaMP expression. The authors underline that most errors were false-negatives, which have a milder impact in terms of result interpretation, but the rate of false positives was, nevertheless, relevant in detecting a second fluorophore from a cell expressing only one protein. The experimental profiles of fluorophores were dependent both on the specific fluorescent protein and on the projecting area, and the distribution of double-labelled did not match anatomical evidence. This result should be taken as the limitation of the present pioneering experiments, presented as proof-of-principle of the approach, but Neuroplex may provide far improved precision under different experimental conditions.

In my view, the work of Phillips et al. represents a significant advance in the state-of-the-art of the field. The rigorous analysis of limitations in the use of Neuroplex must be considered an important guideline for future uses of this approach.

Comments on revision:

The authors have adequately addressed my comments.

---

## [Referee Report · Reviewer #2 (Public review)]

Summary:

The manuscript introduces Neuroplex, a pipeline that integrates miniscope Ca²⁺ imaging in freely moving mice with multiplexed confocal and spectral imaging to infer projection identities of recorded neurons. This technical approach is promising and could broaden access to projection-resolved population imaging. However, the core quantitative analyses apply a winner-take-all single-label assignment per neuron even when multiple fluorophores exceed threshold, with additional labels treated descriptively as "secondary hits." While the authors acknowledge and simulate dual labeling, the extent to which this single-label decision rule affects subtype fractions and behavioural comparisons remains uncertain without a multi-label (or probabilistic) sensitivity analysis and propagation of classification uncertainty.

Strengths:

(1) Conceptual advance and practicality: Decoupling acquisition from identity readout constitutes an innovative approach that is, in principle, applicable in laboratories currently using single-color miniscopes.

(2) Engineering thoroughness: The manuscript offers detailed consideration of GRIN optics, spectral libraries, registration procedures, and simulations that address signal-to-noise ratio, background, and class imbalances.

(3) Immediate community value: If demonstrated to be robust, the pipeline could enable projection-resolved analyses without reliance on specialized multicolor miniscopes.

Comments on revision:

The authors have addressed my comments, and I have no further remarks.

---

## [Author Response]

The following is the authors’ response to the original reviews.

**Reviewer #1 (Public review):**
Genetically encoded fluorescent proteins expressed in specific cell types allow recognising them in vivo and, if the protein is a functional indicator, as in the case of genetically encoded calcium indicators (GECIs), to record activity from the same cellular ensemble. Ideally, if proteins (fluorophores) have perfectly distinct spectral properties, signals can be distinguished from as many cell types as the number of employed fluorophores. In practice, fluorescent proteins have non-negligible crosstalk both in absorption and emission bands. In addition, fluorescence contribution of each fluorophore normally varies from cell to cell and therefore spectral properties of cells expressing two or more proteins are different. The work of Phillips et al. addresses this challenge. The authors present an approach defined as "Neuroplex", allowing identification of up to nine cell types from the same number of fluorophores. The fingerprint of each cell is then associated with functional fluorescence from the GECI GCaMP, allowing recording calcium activity from that specific cell. The method is implemented in vivo using head-mounted miniscopes.The authors used a mouse line expressing GCaMP in cortical pyramidal neurons and developed an experimental pipeline. First, they injected the nine AAV viruses, causing expression of fluorophores in a different brain area. The idea was not to image that area, but a non-infected medial prefrontal cortex (mPFC) section where neurons could be infected by their axons projecting in an injected area, in this way being identified by their targeting region(s). A GRIN lens, allowing spectral analysis, was mounted in the mPFC section, and GCaMP fluorescence was then recorded during behavioural tasks and analysed to identify regions of interest (ROIs) corresponding to neuron somata. After functional imaging, the head of the mouse was fixed, spectral analysis was performed, and after necessary correction for chromatic distortions, the fluorophore contribution was determined for each ROI (neuron) from where GCaMP signals were detected. Notably, the procedures for estimation and correction of chromatic aberration and light transmission (described in Figure 2) were a major challenge in their technical achievements. The selection of the nine fluorophores was another big effort. This was done by combining computer simulations and direct measurement of spectra from individual proteins expressed in HEK293 cells. It is important to say that the authors could simulate arbitrary combinations of two or more different fluorophores and evaluate the ability of their algorithm to detect the correct proteins against wrong estimations of false-negative (absence of an expressed protein) or false-positive (presence of a non-expressed protein). Not surprisingly, this ability decreases with the level of GCaMP expression. The authors underline that most errors were false-negatives, which have a milder impact in terms of result interpretation, but the rate of false positives was, nevertheless, relevant in detecting a second fluorophore from a cell expressing only one protein. The experimental profiles of fluorophores were dependent both on the specific fluorescent protein and on the projecting area, and the distribution of double-labelled did not match anatomical evidence. This result should be taken as the limitation of the present pioneering experiments, presented as proof-of-principle of the approach, but Neuroplex may provide far improved precision under different experimental conditions.In my view, the work of Phillips et al. represents a significant advance in the state-of-the-art of the field. The rigorous analysis of limitations in the use of Neuroplex must be considered an important guideline for future uses of this approach.

We appreciate the reviewer’s positive evaluation and thoughtful comments.

**Reviewer #2 (Public review):**
Summary:The manuscript introduces Neuroplex, a pipeline that integrates miniscope Ca²⁺ imaging in freely moving mice with multiplexed confocal and spectral imaging to infer projection identities of recorded neurons. This technical approach is promising and could broaden access to projection-resolved population imaging. However, the core quantitative analyses apply a winner-take-all single-label assignment per neuron even when multiple fluorophores exceed threshold, with additional labels treated descriptively as "secondary hits." While the authors acknowledge and simulate dual labeling, the extent to which this single-label decision rule affects subtype fractions and behavioural comparisons remains uncertain without a multi-label (or probabilistic) sensitivity analysis and propagation of classification uncertainty.

We thank Reviewer #2 for the careful statistical perspective and focus on assignment strategy and uncertainty. Importantly, we emphasize that Neuroplex is presented as a methodological proof-of-principle, not as a definitive quantification of projection convergence.

Strengths:(1) Conceptual advance and practicality: Decoupling acquisition from identity readout constitutes an innovative approach that is, in principle, applicable in laboratories currently using single-color miniscopes.(2) Engineering thoroughness: The manuscript offers detailed consideration of GRIN optics, spectral libraries, registration procedures, and simulations that address signal-to-noise ratio, background, and class imbalances.(3) Immediate community value: If demonstrated to be robust, the pipeline could enable projection-resolved analyses without reliance on specialized multicolor miniscopes.Weaknesses:(1) Single-label assignment in the main analyses: When multiple fluorophores exceed threshold for a neuron/ROI, the workflow applies a winner-take-all rule and assigns a single label (the fluorophore with the largest standardized beta), while additional above-threshold fluorophores are retained only as "secondary hits." This is a reasonable specificity-first choice, but because cortical excitatory neurons can collateralize, collapsing dual-threshold ROIs to one identity may under-represent dual-projecting cells and could bias estimated subtype fractions and behavioural comparisons.

We thank the reviewer for raising this important conceptual point.

We agree that cortical excitatory neurons frequently collateralize and therefore may legitimately express more than one retrograde fluorophore. Our use of a winner-take-all (WTA) rule in the primary analyses was an intentionally conservative methodological choice designed to prioritize specificity over sensitivity in this proof-of-principle study.

As demonstrated in our simulations (Supp. Fig. 5–6), under realistic background and noise conditions, secondary assignments are more susceptible to false-positive errors than primary assignments. For this reason, we chose to assign a single primary identity for quantitative behavioral stratification while retaining additional above-threshold fluorophores as “secondary hits” and reporting their distribution separately (Supp. Fig. 7).

We did not intend to imply that projections are exclusive. Rather, the WTA strategy provides a conservative lower-bound estimate of subtype proportions and avoids inflation of dual-label rates under conditions where spectral separability is imperfect.

We agree that this rationale should be stated more explicitly in the manuscript, and that the potential impact of assignment strategy on subtype fractions and behavioral comparisons should be acknowledged clearly as a methodological trade-off rather than a biological claim.

Importantly, the biological analyses presented in this manuscript are illustrative demonstrations of functional stratification capability and do not depend on exclusivity of projection identity. We have revised the manuscript to clarify this framing as follows:

“If multiple fluorophores exceeded the threshold for an ROI, the fluorophore with the largest z-scored beta value was assigned as the primary identity (winner-take-all rule). This conservative approach was chosen to prioritize specificity under realistic noise and background conditions. Additional above-threshold fluorophores were retained as ‘secondary hits’ but were not incorporated into primary subtype stratification analyses.” (Methods, Single Pass Algorithm)

“For quantitative behavioral comparisons, each ROI was assigned a single primary fluorophore identity using a winner-take-all rule. We emphasize that this assignment strategy does not imply projection exclusivity. Rather, it provides a conservative lower-bound estimate of subtype proportions, as ROIs exceeding threshold for multiple fluorophores were classified according to their strongest spectral contribution.” (Result, Fluorophore distribution in behaviorally relevant ROIs)

“These analyses were performed using conservative single-label assignments; dual-threshold ROIs were not treated as co-identities in order to avoid overinterpretation of potentially ambiguous multi-label cells. Because identity assignment prioritizes specificity and classification uncertainty was not formally propagated into downstream comparisons, subtype fractions and behavior-by-subtype differences should be interpreted as qualitative demonstrations of projection-resolved functional stratification rather than precise anatomical quantifications. ” (Results, Neuronal Cell Type and Behavior)

“Cortical pyramidal neurons frequently collateralize to multiple downstream targets, and accordingly some ROIs exceeded threshold for more than one fluorophore. In this proof-of-principle implementation, we adopted a specificity-first winner-take-all assignment rule for primary analyses to minimize false-positive multi-label calls under realistic noise conditions. This strategy likely underestimates the true prevalence of dual-projecting neurons and should therefore be interpreted as a conservative stratification approach rather than a statement of projection exclusivity.” (Discussion)

(2) Dual-label detection is acknowledged but remains descriptive in vivo: the manuscript explicitly discusses the possibility of dual projection, evaluates dual-fluorophore detection in simulations (including performance under realistic noise/background), and reports in vivo rates of secondary hits. However, these dual-threshold events are not incorporated as co-identities in the main statistical analyses, making it difficult to judge how robust the principal biological conclusions are to the single-label decision rule.

We thank the reviewer for this important clarification request.

We agree that dual-projection neurons are biologically plausible and that dual-threshold ROIs were detected in vivo. In this manuscript, however, our primary goal was to establish the feasibility of high-dimensional spectral assignment and projection-resolved stratification, rather than to provide a definitive quantification of projection convergence.

For this proof-of-principle study, we chose a conservative winner-take-all (WTA) framework for primary behavioral analyses in order to minimize false-positive multi-label assignments under realistic noise and background conditions, as demonstrated in our simulations (Supp. Fig. 5–6). Secondary hits were retained and reported descriptively (Supp. Fig. 7), but not incorporated into the primary statistical comparisons to avoid overinterpretation of potentially ambiguous dual-label calls.

Importantly, the principal biological conclusions presented in the manuscript are qualitative demonstrations that projection-defined stratification is feasible within a single animal. These conclusions do not rely on projection exclusivity or on precise quantification of dual-projecting fractions.

We agree that this distinction should be made clearer in the manuscript, and we have revised the text as follows:

“Although dual-threshold ROIs were detected in vivo, these secondary assignments were not incorporated as co-identities in the primary behavioral analyses. This decision reflects a conservative specificity-first framework designed to minimize false-positive multi-label calls under realistic noise conditions. Accordingly, dual-label rates reported here should be interpreted descriptively. The present study focuses on demonstrating the feasibility of projection-resolved stratification, rather than providing definitive quantification of projection convergence.” (Results, Fluorophore distribution in behaviorally relevant ROIs)

“We then stratified these neurons by projection target and examined behaviorally selective activity across cell types. These analyses were performed using conservative single-label assignments; dual-threshold ROIs were not treated as co-identities in order to avoid overinterpretation of potentially ambiguous multi-label cells. Because identity assignment prioritizes specificity and classification uncertainty was not formally propagated into downstream comparisons, subtype fractions and behavior-by-subtype differences should be interpreted as qualitative demonstrations of projection-resolved functional stratification rather than precise anatomical quantifications.” (Results, Behavioral Analysis)

(3) Uncertainty is not propagated: False-positive/false-negative rates from simulations and uncertainty from registration/segmentation are not carried forward into quantitative confidence bounds on subtype proportions or behaviour-by-subtype effects.

We agree that formal propagation of classification and registration uncertainty into subtype proportions and behavioral comparisons would be appropriate in a study primarily focused on precise anatomical quantification. However, the central goal of the present manuscript is methodological and to demonstrate that high-dimensional spectral identity can be reliably linked to miniscope-recorded functional activity within a single animal.

We have shown that simulations under realistic noise, background, and class imbalance conditions (Supp. Fig 5-6) show that errors are predominantly false negatives rather than false positives. However, behavioral analyses are presented as qualitative demonstrations of the feasibility of projection-resolved stratification rather than as definitive quantitative anatomical measurements.

In the revised manuscript, we clarified that (1) subtype proportions and behavioral effects are assignment-dependent estimates, (2) simulation-derived error rates provide guidance for experimental design rather than formal confidence intervals, and (3) future studies centered on precise quantification of projection fractions would benefit from formal uncertainty modeling, as follows:

“These simulation-derived accuracy estimates characterize expected performance under defined noise and background conditions but were not formally propagated into confidence bounds on subtype proportions or behavioral comparisons. In this proof-of-principle study, subtype fractions are presented as assignment-dependent estimates rather than definitive anatomical measurements.” (Results, Assessment of spectral unmixing approach)

“Because classification uncertainty was not formally propagated into these analyses, behavior-by-subtype comparisons should be interpreted as qualitative demonstrations of functional stratification rather than precise quantitative estimates.” (Results, Neuronal cell types and behavior)

“The modeling framework was designed to characterize expected classification behavior across a range of experimental regimes, including background fluorescence, class imbalance, and reduced signal-to-noise ratio. These simulations provide practical performance guidance but were not used to compute formal error bars or propagate uncertainty into downstream biological analyses.” (Methods, Modeling of experimental variables to assess accuracy of algorithms)

“Because the present study is designed to establish methodological feasibility rather than precise anatomical quantification, simulation-derived false-positive and false-negative regimes were not formally propagated into confidence bounds on subtype proportions or behavioral effect sizes. Accordingly, subtype fractions should be interpreted as assignment-dependent estimates rather than definitive anatomical measurements. Future implementations could incorporate Bayesian or likelihood-based classifiers to generate posterior identity probabilities and enable formal uncertainty propagation when quantitative estimation of projection convergence is central to the biological question.” (Discussion)

**Reviewer #3 (Public review):**
This manuscript presents Neuroplex, a technically rigorous and carefully validated pipeline that links miniscope calcium imaging in freely behaving animals with high-dimensional fluorophore-based cell-type identification using in vivo multiplexed spectral confocal imaging through the same implanted GRIN lens. The work overcomes a major practical limitation of head-mounted microscopy by enabling the identification of up to nine projection-defined neuronal populations within the same animal, without post-fixation histology. The approach is well motivated and supported by extensive calibration and simulation. While the biological results are primarily illustrative, the methodological contribution is clear and likely to be broadly useful.Major comments(1) The approach relies on the assumption that fluorophore identity assigned during anesthetized confocal imaging accurately reflects the identity of neurons recorded during prior behavioural sessions. While the use of the same GRIN lens and in vivo co-registration mitigates many concerns, the manuscript would benefit from a more explicit discussion, or empirical demonstration, if available, of the stability of fluorophore assignments across time. Even limited repeat spectral imaging in a subset of animals would strengthen confidence in longitudinal applicability.

We thank the reviewer for highlighting this important conceptual assumption.

Fluorophore identity in Neuroplex is genetically encoded via AAVretro delivery and therefore does not depend on transient physiological state. Spectral imaging is performed in vivo through the same GRIN lens and field of view used during behavioral imaging, and co-registration relies on anatomical landmarks. While repeat spectral imaging was not formally performed as a longitudinal experiment, the underlying fluorescent protein expression is stable over weeks, and there is no biological mechanism in this paradigm that would alter fluorophore identity across sessions.

We revised the manuscript to explicitly state this assumption and clarify why identity stability is expected as follows:

“…fluorophore signals and reduce unmixing fidelity, leading to an increased false positive rate. Fluorophore identity in this framework is genetically encoded via retrograde AAV delivery and is therefore expected to remain stable across behavioral and spectral imaging sessions. Because both functional and spectral data are acquired in vivo through the same GRIN lens and co-registered using anatomical landmarks, assignment stability is not expected to vary across time unless expression levels change substantially. While repeat spectral imaging was not performed as a formal longitudinal experiment in this study, the stability of fluorescent protein expression supports the assumption that fluorophore identity reflects a persistent cellular attribute.” (Discussion)

(2) Fluorophore identity is determined using thresholding of linear unmixing coefficients relative to an empirically defined baseline, followed by a second adaptive pass for over-represented fluorophores. While this heuristic is extensively validated via simulations, it remains ad hoc from a statistical perspective. The authors should more explicitly justify this choice and discuss its limitations relative to probabilistic or likelihood-based classifiers, particularly with respect to uncertainty estimation at the single-ROI level.

We agree that the dual-pass thresholding approach is heuristic rather than fully probabilistic. More formal probabilistic classifiers are possible but would introduce additional modeling assumptions and training requirements beyond the scope of this proof-of-principle study.

We revised our manuscript to clarify this as follows:

“The current classification framework relies on linear unmixing followed by empirically defined thresholding rather than full probabilistic inference. This approach provides transparency and practical robustness under realistic noise and background conditions but does not generate single-ROI posterior uncertainty estimates. ” (Discussion)

(3) Identifiability of fluorophores is demonstrated empirically, but the manuscript does not explicitly quantify spectral separability (e.g., similarity metrics between basis spectra or conditioning of the unmixing matrix). A brief analysis of spectral independence or sensitivity of beta estimates to noise would provide mathematical reassurance, especially given the reliance on linear regression in a high-dimensional feature space.

We agree that spectral separability is conceptually important. In this manuscript, separability is demonstrated empirically through (1) In vitro fingerprint acquisition under identical optical conditions, (2) simulation under background and noise, and (3) successful in vivo classification across regimes. We did not compute formal matrix conditioning metrics, but we agree that the separability rationale should be described more explicitly. We revised our manuscript as:

“While formal conditioning metrics were not explicitly computed empirical fingerprint acquisition and simulation-based perturbation analyses demonstrate sufficient spectral independence for reliable linear unmixing under the tested regimes.” (Discussion)

(4) The spectral unmixing treats CNMF-derived ROIs as fixed supports. I wonder whether ROI boundaries, neuropil contamination, and partial overlap can introduce structured uncertainty that could bias spectral estimates. If so, the authors should acknowledge this dependency more explicitly and discuss how ROI quality or overlap might influence false negatives or false positives, particularly in densely labelled regions.

We agree that ROI definition influences spectral extraction. Spectral fingerprints are derived by averaging all pixels within the ROI mask, and therefore neuropil contamination, partial ROI overlap, and dense labeling could influence beta estimates. In the revised manuscript, we have acknowledged this dependencies more explicitly.

“Spectral unmixing operates on CNMF-derived ROI masks treated as fixed supports. Accordingly, segmentation quality, neuropil contamination, and partial overlap between neighboring cells can influence extracted spectral fingerprints and may contribute to false negatives or secondary assignments, particularly in densely labeled regions. These structured sources of uncertainty are expected to have the greatest impact under regimes of extreme class imbalance, low fluorophore brightness, strong neuropil signal, or pairing of spectrally overlapping reporters. Use of refined segmentation strategies or nuclear-localized reporters could reduce such structured uncertainty in future implementations.” (Discussion)

(5) The manuscript reports meaningful rates of secondary fluorophore detection, but also nontrivial false-positive rates for secondary labels under realistic conditions. The authors appropriately caution against over-interpretation, but the Discussion should more clearly delineate when dual-label assignments are likely to be biologically interpretable versus methodologically ambiguous, and how experimental design (e.g., fluorophore pairing) should be optimized accordingly.

We agree and will delineate interpretability boundaries explicitly.

“Dual-label assignments are most reliable when fluorophores are spectrally well separated and when signal-to-noise ratios are high. In contrast, spectrally adjacent fluorophore pairs or densely labeled regimes increase ambiguity and false-positive risk. Experimental design should therefore prioritize pairing spectrally distant fluorophores when projection convergence is of primary interest.” (Discussion)

(6) I suspect that Neuroplex will be most effective in certain regimes (moderate convergence, bright and spectrally distinct fluorophores) and less reliable in others. A more explicit discussion of best practices, anticipated failure modes, and experimental scenarios where the method may be inappropriate would increase the practical value of the paper for adopters.

“More broadly, Neuroplex is expected to perform most robustly in regimes characterized by moderate projection convergence, balanced fluorophore representation, bright and spectrally distinct reporters, and adequate signal-to-noise ratio. Imaging directly within a projection target that has received dense retrograde labeling may introduce substantial class imbalance, which simulations predict will reduce detection sensitivity for the dominant fluorophore. In such cases, conservative assignment strategies, reduced spectral complexity, or refinement of ROI definition may improve interpretability. Careful fluorophore selection and pilot validation under intended imaging conditions are therefore recommended prior to large-scale application. Future implementations incorporating nuclear-localized reporters may further reduce segmentation-dependent ambiguity by constraining spectral signals to somatic compartments.”

**Recommendations for the authors:**

**Reviewer #1 (Recommendations for the authors):**
The authors should address a few points that are not clear.(1) At the end of the Results, the authors assess their approach using only four fluorophores and conclude that Neuroplex works "even" under reduced complexity. There is something I am missing. In my mind, lower complexity should be easier and should work better. As a researcher, I would first assess a four-fluorophores scenario and then step up with complexity, but the authors did the opposite. Also, I think that the present Supplementary Figure 9 should be in the main text; I don't understand why the authors decided to relegate a clear result to the bottom of everything. The authors should give some explanations.

We agree that reduced spectral complexity should, in principle, improve separability and classification performance. Our original presentation order was intended to first demonstrate feasibility under the most challenging condition (nine fluorophores plus GCaMP), thereby establishing maximal multiplexing capacity. The reduced-complexity experiment was included to demonstrate scalability and generalizability under more typical experimental regimes. However, we agree that this rationale was not sufficiently clear and that the reduced-complexity results merit presentation in the main text.

Accordingly:

We have moved former Supplementary Figure 9 into the main Results (Fig. 6).

We have clarified explicitly why the nine-fluorophore condition was presented first as follows:

“To evaluate the performance of Neuroplex under more typical experimental regimes with reduced-complexity, we applied the pipeline to two GCaMP transgenic animals injected with a subset of four fluorophores.”

(2) The question of relative expression is crucial. Among the infected regions, there is the contralateral mPFC and I imagine that if they image there, the contribution of the expressed protein might dominate all other components, preventing detection of other fluorophores, including GCaMP. But is it the case, or would it be possible to detect projecting neurons in that region? I would be surprised that the authors never tried it; this test would simply imply mounting the GRID lens on the other hemisphere.

This is an important conceptual point.

Our simulations (Supp. Fig. 5) explicitly model over-representation of a single fluorophore. These results show that heavy class imbalance primarily increases false negatives (due to baseline normalization) rather than false positives.

In the revised manuiscript, we discussed this limitation more explicitly.

“Relative fluorophore representation within the imaged field of view influences classification robustness. As demonstrated in our simulations of class imbalance (Supp. Fig. 5g–h), extreme over-representation of a single fluorophore primarily increases false-negative rates due to baseline normalization effects. In the present study, we intentionally avoided imaging directly within heavily infected projection targets (e.g., contralateral mPFC) in order to maintain moderate fluorophore representation across ROIs. Imaging in a densely labeled region would represent a more challenging regime, and we would expect reduced sensitivity for the dominant fluorophore under such conditions.” (Dicussion)

(3) The possibility to utilise Neuroplex goes beyond the type of experiment presented as proof-of-concept in this technical paper. In the Discussion, the authors mention genetically defined subtypes and activity-tagged neurons. But, if one changes the pipeline, can it be used by expressing GECIs with different spectra, or GECIs and genetically-encoded voltage indicators (GEVIs)? I would be very interested in knowing what the authors think about this putative "shortcut".

We thank the reviewer for this forward-looking and insightful question.

In principle, the Neuroplex framework could be extended to incorporate spectrally distinct genetically encoded functional indicators, including multi-color GECIs or combinations of GECIs and GEVIs. However, it is important to distinguish this from the identity-assignment strategy implemented in the present study.

Simultaneous multi-color functional imaging under a head-mounted miniscope is optically more demanding than assigning cell identity from single-color functional recordings followed by high-dimensional spectral readout. Multi-color GECI or GEVI imaging requires real-time excitation and emission separation during dynamic recording, increases optical complexity, and is particularly sensitive to chromatic aberration, photon efficiency, and signal-to-noise constraints imposed by GRIN lenses.

In contrast, Neuroplex decouples functional acquisition from spectral identity determination. Functional activity is recorded using a single optimized channel, while spectral separation is performed separately under controlled confocal conditions with multiplexed excitation and emission sampling. This design substantially reduces optical burden during behavioral imaging.

While integration of multiple functional reporters is conceptually feasible within this framework, successful implementation would require careful validation of brightness, spectral separability, and temporal stability for each reporter combination.

**Reviewer #2 (Recommendations for the authors):**
(1) Implement a principled multi-label calling mode for cells with >1 above-threshold fluorophore (e.g., per-fluorophore FDR control or Bayesian posteriors). Report cell-wise weights and re-run key results three ways: single-label, hard multi-label, and soft (probabilistic) assignments; state explicitly how conclusions change.

We appreciate this suggestion and agree that multi-label or probabilistic calling frameworks are well motivated, particularly for studies in which projection convergence is the central biological question. In the current manuscript, however, our goal is to establish a practically deployable proof-of-principle pipeline for linking miniscope functional recordings to a high-dimensional spectral-identity readout. Consistent with this scope, we used a conservative winner-take-all (WTA) strategy for primary analyses to prioritize specificity under realistic noise and background conditions, and we treated multi-hit events descriptively. Importantly, the qualitative conclusions regarding projection-resolved functional stratification are unchanged when secondary-hit distributions are examined.

In the revised manuscript, we explicitly stated that: (i) single-label assignment is a conservative analysis choice rather than a biological claim of exclusivity, and (ii) multi-label or probabilistic calling is a natural extension for future work, as follows:

“If multiple fluorophores exceeded the threshold for an ROI, the fluorophore with the largest z-scored beta value was assigned as the primary identity (winner-take-all rule). This conservative approach was chosen to prioritize specificity under realistic noise and background conditions. Additional above-threshold fluorophores were retained as ‘secondary hits’ but were not incorporated into primary subtype stratification analyses.” (Methods, Single Pass Algorithm)

“Because the present study is designed to establish methodological feasibility rather than precise anatomical quantification, simulation-derived false-positive and false-negative regimes were not formally propagated into confidence bounds on subtype proportions or behavioral effect sizes. Accordingly, subtype fractions should be interpreted as assignment-dependent estimates rather than definitive anatomical measurements. Future implementations could incorporate Bayesian or likelihood-based classifiers to generate posterior identity probabilities and enable formal uncertainty propagation when quantitative estimation of projection convergence is central to the biological question.” (Discussion)

(2) Add ground truth for dual projectors in a subset (paired orthogonal tracers or staged injections) and provide a confusion matrix including dual-positives; use this to calibrate thresholds/priors.

We agree that ground truth validation of dual projectors using orthogonal tracers or staged injections would be valuable, particularly for calibrating priors and enabling confusion-matrix-based evaluation. However, these experiments require additional cohorts and experimental design beyond the scope of the current proof-of-principle technical manuscript. Our goal here is to demonstrate the feasibility of multiplexed identification and projection-resolved stratification within a single animal, not to provide definitive anatomical quantification of collateralization.

We have revised the manuscript to clearly state that dual-label in vivo observations are descriptive and that studies aimed at quantitative convergence mapping should incorporate orthogonal ground truth validation.

“Accurate quantification of projection convergence would benefit from orthogonal ground-truth validation (e.g., paired tracers or staged injections) to establish confusion matrices for dual positives and to calibrate thresholds or priors.”

(3) Propagate uncertainty from simulations and registration/segmentation to subtype fractions and behavior effects (error bars or sensitivity analyses).

We agree that formal uncertainty propagation is appropriate for studies focused on precisely quantifying subtype proportions or effect sizes. In this manuscript, subtype fractions and behavioral comparisons are presented primarily as demonstrations of the feasibility of projection-resolved functional stratification, rather than definitive anatomical measurements. Simulation analyses are included to characterize expected performance under defined noise and background regimes, but we did not propagate these uncertainties into downstream confidence bounds in this proof-of-principle work.

We have revised the manuscript to clarify this explicitly as follows:

“These simulation-derived accuracy estimates characterize expected performance under defined noise and background conditions but were not formally propagated into confidence bounds on subtype proportions or behavioral comparisons. In this proof-of-principle study, subtype fractions are presented as assignment-dependent estimates rather than definitive anatomical measurements.” (Results, Assessment of spectral unmixing approach)

“These analyses were performed using conservative single-label assignments; dual-threshold ROIs were not treated as co-identities in order to avoid overinterpretation of potentially ambiguous multi-label cells. Because identity assignment prioritizes specificity and classification uncertainty was not formally propagated into downstream comparisons, subtype fractions and behavior-by-subtype differences should be interpreted as qualitative demonstrations of projection-resolved functional stratification rather than precise anatomical quantifications.” (Results, Neuronal cell types and behavior)

“The modeling framework was designed to characterize expected classification behavior across a range of experimental regimes, including background fluorescence, class imbalance, and reduced signal-to-noise ratio. These simulations provide practical performance guidance but were not used to compute formal error bars or propagate uncertainty into downstream biological analyses.” (Methods, Modeling of experimental variables to assess accuracy of algorithms)

“Because the present study is designed to establish methodological feasibility rather than precise anatomical quantification, simulation-derived false-positive and false-negative regimes were not formally propagated into confidence bounds on subtype proportions or behavioral effect sizes. Accordingly, subtype fractions should be interpreted as assignment-dependent estimates rather than definitive anatomical measurements. Future implementations could incorporate Bayesian or likelihood-based classifiers to generate posterior identity probabilities and enable formal uncertainty propagation when quantitative estimation of projection convergence is central to the biological question.” (Discussion)

(4) Mitigate sources of spurious multi-hits (neuropil handling, ROI mask erosion, nuclear-localized reporters, spectral basis choices) and quantify their impact on dual-label recovery.

We agree that neuropil contamination, ROI boundary choices, and spectral basis selection can influence multi-hit rates. In the current manuscript, we already implement background subtraction and evaluate multi-hit behavior through simulations under realistic background and noise regimes. Quantitative evaluation of additional mitigation strategies (e.g., ROI erosion comparisons) would require new analyses beyond the current scope.

We have revised the Discussion to include concrete best-practice recommendations (e.g., fluorophore pairing, conservative interpretation of multi-hits, and potential use of nuclear-localized reporters).

“Multi-hit events can reflect true biological collateralization but may also arise from structured sources of ambiguity such as neuropil contamination, partial ROI overlap, or imperfect ROI boundaries. These factors may bias spectral estimates and contribute to secondary assignments, particularly in densely labeled regions. Practical mitigation strategies include conservative assignment rules, improved segmentation, and use of nuclear-localized reporters to reduce neuropil contribution. ”

(5) Clarify claims in the main text/figures wherever exclusivity is implied; label which panels use single-label vs multi-label/soft assignments.

We agree and thank the reviewer for emphasizing clarity. We did not intend to imply projection exclusivity. We have revised the manuscript text and figure legends to explicitly state where single-label (winner-take-all) assignment is used, and to avoid language that could be read as claiming exclusive projection identity as follows:

“For quantitative behavioral comparisons, each ROI was assigned a single primary fluorophore identity using conservative winner-take-all rule. This assignment reflects the strongest spectral contribution and does not imply projection exclusivity. Rather, it provides a conservative lower-bound estimate of subtype proportions, as ROIs exceeding threshold for multiple fluorophores were classified according to their strongest spectral contribution.”